# Lyn kinase regulates egress of flaviviruses in autophagosome-derived organelles

Ming Yuan Li[1], Trupti Shivaprasad Naik[1], Lewis Yu Lam Siu[1], Oreste Acuto[2], Eric Spooner[3], Peigang Wang [4], Xiaohan Yang[5], Yongping Lin[6], Roberto Bruzzone [1], Joseph Ashour[7,8], Matthew J Evans[7] & Sumana Sanyal [1,2✉]

Among the various host cellular processes that are hijacked by flaviviruses, few mechanisms have been described with regard to viral egress. Here we investigate how flaviviruses exploit Src family kinases (SFKs) for exit from infected cells. We identify Lyn as a critical component for secretion of Dengue and Zika infectious particles and their corresponding virus like particles (VLPs). Pharmacological inhibition or genetic depletion of the SFKs, Lyn in particular, block virus secretion. Lyn$^{-/-}$ cells are impaired in virus release and are rescued when reconstituted with wild-type Lyn, but not a kinase- or palmitoylation-deficient Lyn mutant. We establish that virus particles are secreted in two distinct populations – one as free virions and the other enclosed within membranes. Lyn is critical for the latter, which consists of proteolytically processed, infectious virus progenies within autophagosome-derived vesicles. This process depends on Ulk1, Rab GTPases and SNARE complexes implicated in secretory but not degradative autophagy and occur with significantly faster kinetics than the conventional secretory pathway. Our study reveals a previously undiscovered Lyn-dependent exit route of flaviviruses in LC3+ secretory organelles that enables them to evade circulating antibodies and might affect tissue tropism.

[1] HKU-Pasteur Research Pole, School of Public Health, Li Ka Shing Faculty of Medicine, The University of Hong Kong, Hong Kong, Hong Kong SAR. [2] Sir William Dunn School of Pathology, University of Oxford, South Parks Road, Oxford OX1 3RE, UK. [3] Whitehead Institute for Biomedical Research, Massachusetts Institute of Technology, Nine Cambridge Center, Cambridge, MA 02142, USA. [4] Department of Microbiology, School of Basic Medical Sciences, Capital Medical University, 100069 Beijing, China. [5] Medical Genetic Centre, Guangdong Women and Children Hospital, 511400 Guangdong, China. [6] Department of Laboratory Medicine, The First Affiliated Hospital of Guangzhou Medical University, 510120 Guangdong, China. [7] Department of Microbiology, Icahn School of Medicine at Mount Sinai, New York, NY, USA. [8] Present address: Boehringer Ingelheim, Richmond, CT, USA. ✉email: sumana.sanyal@path.ox.ac.uk

Dengue and Zika represent two of the major mosquito-borne flaviviruses that collectively have huge health implications worldwide[1,2]. Dengue infects >50 million people annually, often causing severe pathologies such as vascular endothelial leakage. Zika too has emerged as a global threat with recent outbreaks linked to serious neurodevelopmental complications in children and Guillain–Barré syndrome in adults[3]. Vaccines and therapeutic options for these viruses are currently unavailable, along with limited knowledge on the underlying mechanisms of pathogenesis and viral manipulation of host cell biology.

Dengue and Zika viruses exhibit significant overlap in their genome organisation, intracellular life cycle and exploitation of host cellular processes[4,5]. Both rely on specific interactions with host factors that are necessary for viral entry, replication, assembly and release[6]. Although several genetic screens have uncovered host dependency factors for viral entry and replication[7], few studies have investigated mechanisms underlying assembly and release of viral progenies. Infection is accompanied by induction of lipophagy[8,9], followed by massive membrane reorganisation and formation of assembly sites that appear as membrane invaginations along the ER. Immature virions bud into the ER lumen, and traffic along the cellular secretory pathway from the ER through the Golgi complex, where they undergo maturation to form infectious particles prior to exiting the cell via as-yet undefined routes.

Current evidence on the mode of secretion and cell-to-cell spread of flaviviruses is limited. For Dengue and Zika, a few mechanisms have been proposed. Secretion of infectious viral RNA has been speculated to occur through exosomal vesicles[10]. Cell-to-cell spread of Dengue virus was reported to be facilitated by tetraspannin CD189 + organelles[11]. More recently, production of mature infectious Dengue virions was reported to depend on cellular autophagy[12]. Autophagy-associated vesicles were also shown to carry infectious Dengue particles that allowed virus transmission while avoiding antibody neutralisation[13]. Similarly, transmission of Zika from placental cells was reported to occur by packaging into exosome-like organelles[14].

Secretory autophagy is a poorly understood process. It was recently described as one of the modes of unconventional secretion where autophagosomes fuse with the plasma membrane instead of lysosomes[15]. Cellular cargo that co-opt this pathway include IL-1β[16], lysozyme in Paneth cells[17] and mitochondria in reticulocytes[18]. Among intracellular pathogens, enteroviruses such as polio, coxsackievirus B[19,20] and enterovirus 71 have been described to exit cells as particle populations via secretory autophagosomes[21].

Interaction between Dengue prM and cellular KDEL receptors (KDELRs) facilitates ER-to-Golgi transport during secretion of Dengue particles[22]. Since KDELRs have been reported to trigger activation of Src-family kinases akin to G-protein-coupled receptors, we hypothesised that SFK-dependent signalling is also triggered during secretion of viral progenies. We therefore screened for activated SFKs in Dengue-infected cells. Of the nine SFK members[23], we identified three—Lyn, Fyn and Src that were heavily phosphorylated in their activation loop (Y420 for Fyn, Y397 for Lyn and Y419 for Src) during infection. Lyn deficiency in particular resulted in severely impaired secretion of progeny virions and virus-like particles. Using biochemical and imaging analyses, we established that Lyn triggered virus transport in specialised organelles that were LC3+, indicative of autophagosomes. Separation of extracellular virus particles revealed distinct populations of non-infectious free virions and membrane-enclosed, highly infectious particles. Lyn deletion resulted in loss of the membrane-enclosed, infectious particles. This effect could be recapitulated by systematic depletion of genes implicated in secretory, but not degradative autophagy. Lyn-dependent

transport in autophagosomes was triggered specifically by processed, mature virions, but not by furin-resistant immature particles. The mechanism of flavivirus secretion has long been speculated; our study uncovers a Lyn-dependent exit strategy triggered by flaviviruses that might enable them to evade circulating antibodies and dictate tissue tropism.

## Results

**Flavivirus infection activates Src-family kinases.** A pulse of protein flux from the ER to Golgi compartments is known to activate a KDELR-Src-dependent signalling cascade that is necessary to maintain inter-organellar trafficking[24]. Since Dengue progeny virions bind to KDELRs to arrive at the Golgi, we hypothesised that an analogous SFK-dependent signalling is likely triggered during infection to enable virus transport. To test this hypothesis, we measured activation of SFKs in mock- and Dengue virus-infected cells, 24 h post infection, using antibodies that specifically recognise phosphorylation at their activation loop. SFKs displayed >fivefold increase in their phosphorylation status in infected cells measured at varying MOI (0.1, 1, 5 or 10) in two different cell lines (BHK21 and Vero) (Fig. 1a, b). We verified this phenomenon in Zika virus-infected cells, which is a closely related flavivirus. Phosphorylation of SFKs increased both in a time- and dose-dependent manner (Fig. 1c, d).

To focus exclusively on virus release, we took advantage of a VLP-secretion model. These cells stably express the viral E and prM proteins (from Dengue serotypes 1–4, or Zika), and constitutively secrete VLPs, thus mimicking virus budding, maturation and release[25]. Here too, phosphorylation of SFKs was significantly enhanced (>5-fold) in cells secreting VLPs for all Dengue serotypes as well as Zika, as compared to their parental control cells (Fig. 1e, f). As positive control for SFK activation, cells were transfected with a previously characterised HRP construct carrying a KDEL sequence (ssHRP$^{KDEL}$)[24], which displayed similar levels of phosphorylated SFKs as virus-infected cells. The interaction of prME with KDELR was sufficient to activate SFKs and was abolished in the presence of a mutant prME that does not bind KDELR as previously described[22] (Supplementary Fig. 1a, b). These results support our hypothesis that intracellular transport of progeny virions or VLPs, irrespective of their serotype, triggers activation of cellular SFKs.

**Identification of SFKs activated during flavivirus infection.** There are at least nine members of SFKs expressed in different combinations in all mammalian cells[23]. To identify those that are activated upon Dengue infection, we immunoprecipitated phosphorylated SFKs using anti-pSFK antibodies from cell lysates prepared from mock- or Dengue-infected BHK21/Vero E6 cells. Phosphorylated SFKs and the associated cellular factors were isolated, resolved by SDS-PAGE and detected by silver staining (Fig. 2a). The lanes were sliced into 2-mm sections and subjected to trypsin digest for identification by mass spectrometry. We identified three members—Src, Fyn and Lyn, and several SFK-regulated substrates in Dengue-infected samples (Fig. 2b). Among the co-immunoprecipitating proteins, several were components of the secretory pathway, ER/Golgi-resident proteins and vesicular transport machinery, including the KDELRs, which we had previously characterised as important for Dengue secretion[22].

The three SFKs displayed high levels of expression in Dengue-susceptible cell lines as confirmed by immunoblotting (Fig. 2c). To confirm their activation, we immunoprecipitated phosphorylated proteins from mock- and Dengue virus-infected Vero cells using anti-p-tyrosine antibodies. Eluates from immunoprecipitated material were analysed by western blotting with specific antibodies against the selected kinases—Src, Fyn and Lyn

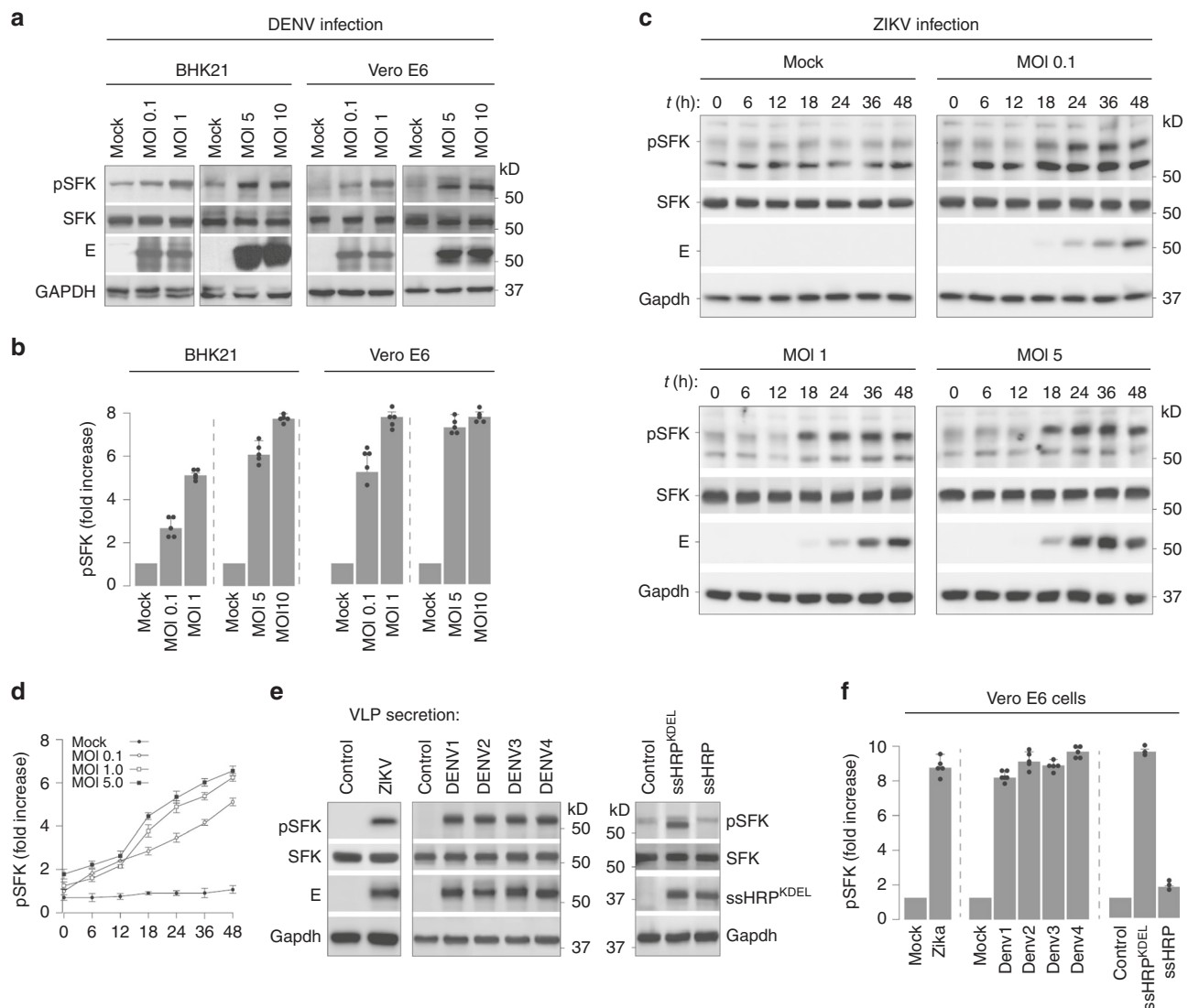

**Fig. 1 Flavivirus infection triggers activation of Src-family kinases.** **a** BHK21 and Vero E6 cells were infected with Dengue virus at varying MOI (0.1, 1.0, 5 and 10) for 24 h. Lysates prepared from infected cells were immunoblotted with antibodies against phosphorylated SFKs, total SFK and anti-flavivirus envelope 4G2 antibodies to detect virus particles. Gapdh was used as loading control. **b** Densitometric analyses were performed on immunoblots. Data are presented as fold change of pSFK expression normalised to total SFK compared to mock-infected cells (mean ± s.d of five independent experiments). **c** Cells were infected with Zika virus at varying MOI of 0.1, 1 and 5. At indicated time intervals post infection, lysates prepared from mock and infected cells were immunoblotted with anti-pSFK antibodies, anti-SFK and 4G2 antibodies to detect virions. Gapdh levels were measured as loading control. **d** Densitometric analyses were performed on immunoblots from four independent experiments. Data are presented as fold change of pSFK/total SFK of each sample compared to mock-infected cells set at 1. Error bars represent mean ± s.d. from four independent experiments. **e** Cells constitutively secreting either Zika (left panel) or Dengue (middle panel) VLPs were immunoblotted to detect phosphorylated SFKs and total SFK. Cells expressing the secretory reporter proteins ssHRP$^{KDEL}$ or ssHRP were measured as positive and negative controls for SFK activation (right panel). **f** Densitometric analyses were performed on immunoblots from five to six independent experiments (Zika and Dengue) and three independent experiments for ssHRP. Data are presented as fold change of pSFK/total SFK for each sample compared to control cells set at 1. Error bars represent mean ± s.d. from five independent experiments (Zika and Dengue) and three independent experiments (ssHRP).

(Fig. 2d). Although expression levels of total SFKs were comparable in mock and infected samples, a significant increase was noted in their phosphorylated form in eluates from Dengue virus-infected samples compared to mock (Fig. 2d). We also employed a reciprocal strategy where kinases from mock- and Dengue-infected samples were first immunoprecipitated on specific antibodies followed by immunoblotting with anti-phospho-tyrosine antibodies, which confirmed these results (Fig. 2e). The Dengue structural E protein however, did not co-purify with phosphorylated SFKs, suggesting that they do not directly interact with each other. To further quantitate increases

in specific SFK activation, we took advantage of the Milliplex Map 8-plex SFK activation kit using the Luminex technology, which confirmed specific activation of Lyn, Src and Fyn in lysates prepared from Dengue- and Zika-infected cells (Fig. 2f).

To visualise the subcellular distribution of pSFKs upon virus infection, we performed confocal imaging in Dengue- (Fig. 2g, h) and Zika-infected (Fig. 2i, j) cells. While in mock-infected cells, SFKs appeared to be confined to the Golgi compartments, upon virus infection, the distribution spread throughout the secretory pathway and also in LC3+ compartments, displaying significant colocalisation with viral E protein (Fig. 2g–j).

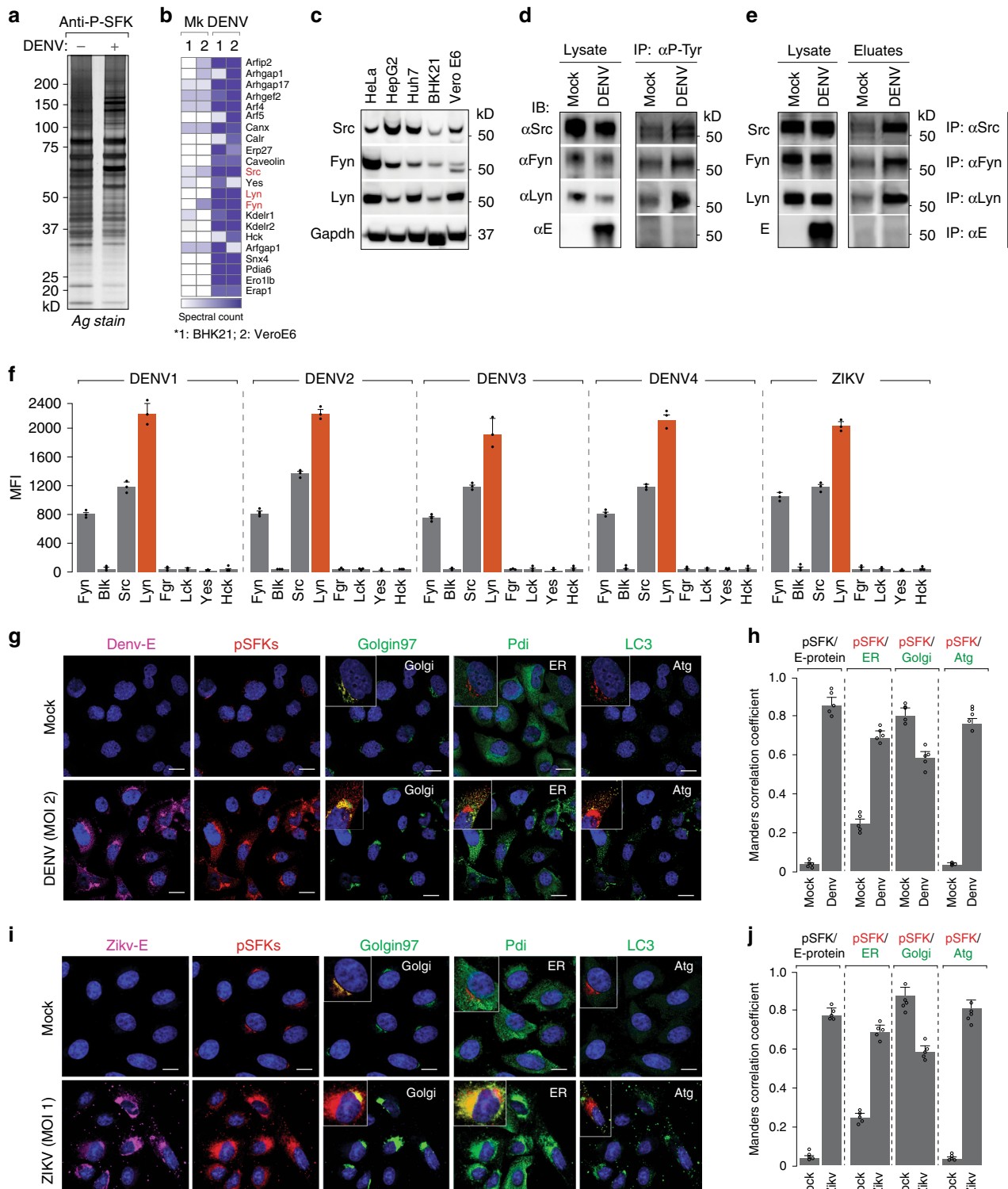

**SFKs facilitate flavivirus production**. To study the role of selected SFKs in flavivirus infection, particularly in viral transport along the secretory pathway, we inhibited SFK activity using two approaches. First, we selected a commercially available SFK inhibitor—SU6656[26], which specifically blocks their activation as measured by the absence of their phosphorylated forms (Supplementary Fig. 2a). We performed MTT assays to establish that at concentrations of SU6656 < 10 μM, cell viability was not affected (Supplementary Fig. 2b). We selected this concentration range to test virus production upon SU6656 treatment. Our

results indicated that 5 μM SU6656 was sufficient to cause ~10-fold reduction in viral titres (Supplementary Fig. 2c). SU6656 treatment did not affect replication efficiency or intracellular vRNA levels (Supplementary Fig. 2d, e). However, we detected a significant loss in extracellular vRNA levels (~10-fold) and in secreted VLPs (>50%) upon SU6656 treatment (Supplementary Fig. 2f–h), accompanied by redistribution of viral prME to lysosomes (Supplementary Fig. 2i, j).

To genetically deplete SFKs, we transfected siRNA targeting the individual kinases and measured both viral titres, and secretion of

**Fig. 2 Flavivirus infection triggers activation of three specific SFKs. a** Large-scale immunoprecipitation of activated SFKs was performed on anti-phospho-SFK antibodies from mock- and Dengue-infected BHK21 and Vero E6 cells (MOI 2, 24 h). Isolated proteins were resolved by SDS-PAGE and detected by silver staining. **b** Entire lanes on gels were sliced into 2-mm sections and subjected to trypsin digest. The peptide mix was processed and analysed by an LTQ Orbitrap mass spectrometer. **c** Protein expressions of Lyn, Fyn and Src kinases were validated in lysates prepared from indicated Dengue-susceptible cell types. **d** Activation of Lyn, Fyn and Src upon Dengue infection was measured by immunoprecipitating first on anti-phospho-tyrosine antibodies and immunoblotting with specific SFK antibodies. **e** Reciprocal immunoprecipitation on specific anti-SFK antibodies followed by immunoblotting with anti-phospho-tyrosine antibodies. **f** Activation of SFKs was measured in lysates prepared from Dengue-infected cells with the Milliplex MAP 8-plex assay kit using the Luminex system as read-out, following the manufacturer's protocol. Red bars indicate Lyn activation. Error bars represent mean ± s.d. from three biological replicates. **g** Colocalisation of Dengue and SFKs was visualised by 4G2 antibodies and anti-phospho-SFKs with markers for ER (Pdi), Golgi (Golgin97) and autophagosomes (LC3) as controls. Insets indicate colocalisation of markers with pSFKs. Images are representative of five independent experiments. Scale bars represent 20 μm. **h** Manders correlation coefficients were measured for colocalisation of pSFKs with E protein, Golgi, ER and autophagosome marker proteins from 50 cells per condition (error bars represent mean ± s.e.m. of five independent experiments). **i**, **j** Same as **g**, **h** in Zika-infected cells (error bars represent mean ± s.e.m. of five independent experiments).

VLPs (Supplementary Fig. 3a–e). Immunoblots with selective antibodies indicated significant reduction in protein expression of targeted SFKs, without altering that of the others (Supplementary Fig. 3a). Among the three kinases, depletion of Lyn had the most significant effect, with ≥10-fold reduction in viral titres for all strains, with no apparent effect on cell viability or intracellular vRNA levels (Supplementary Fig. 3b–d). These data were reflected in the VLP-secretion assay, where a similar reduction was noted in Lyn-depleted cells (Supplementary Fig. 3e). A combined depletion of Lyn with either Fyn or Src displayed an additive effect on inhibition of VLP secretion (Supplementary Fig. 3f, g). Collectively, these results suggest that Lyn in particular plays a critical role in regulating secretion of viral progenies.

**Secretion of viral progenies is dependent on Lyn kinase.** To confirm the role of Lyn in flavivirus infections, we generated a range of Lyn-deleted cell lines using the CRISPR/Cas9 technology (Fig. 3a). $Lyn^{-/-}$ cells were also reconstituted with wild-type (rLyn) and a mutant variant of Lyn (C468A) through lentiviral transduction. Cysteine 468 is located at the C terminus of the kinase domain, and the Lyn mutant carrying an alanine substitution at C468 impairs phosphorylation of Lyn in its activation loop, thereby blocking its kinase activity[27]. An additional mutant with a cysteine-to-serine substitution was generated to prevent palmitoylation of Lyn. All the generated cells (wt, $Lyn^{-/-}$, rLyn, C468A, C3S) were then challenged with either the four serotypes of Dengue or Zika virus. We collected supernatants from mock and infected cells to measure viral titres by plaque assay and viral RNA by RT qPCR (Fig. 3b–f). Virus production was significantly attenuated in all $Lyn^{-/-}$ cells, and could be rescued in the rLyn-expressing cells, but not in those expressing the kinase-inactive C468A mutant or the palmitoylation-deficient C3S mutant (Fig. 3b–f). To further characterise the secreted virus particles, we resolved the extracellular particles from [$^{35}$S]cys/met-labelled wt and Lyn-deficient cells by SDS-PAGE and detected by auto-radiography (Fig. 3g). We observed a significant reduction in abundance of extracellular virus particles from Lyn-deficient cells. Secreted particles also displayed a loss in prM cleavage indicative of immature viral progenies (Fig. 3g).

To exclude defects in replication or translation, we measured E-protein synthesis in metabolically labelled cells. The abundance of newly synthesised viral prME was comparable in wild-type and Lyn-deficient cells, indicating that viral protein translation was not affected (Fig. 3h). We also expressed the corresponding viral replicons carrying luciferase reporters in the wt and Lyn-deficient cells and did not detect any significant difference (Fig. 3i).

To verify that attenuated virus production was specifically due to defective secretion, we measured the appearance of VLPs in the supernatants. For both Dengue and Zika, we observed a significant loss in VLP secretion from $Lyn^{-/-}$ cells, which could

be rescued in cells expressing wild-type Lyn (rLyn) but not C468A or C3S mutants, suggesting that both its kinase activity and palmitoylation are necessary for virus release (Fig. 3j, k).

**Virus transport is blocked post Golgi in Lyn-deficient cells.** To determine the underlying mechanism of Lyn-dependent virus transport, first we performed pulse-chase assays to measure viral prME-trafficking characteristics. Mock and infected cells (MOI 2, 24 h) were radiolabelled with [$^{35}$S]cys/met for 1 h, and chased in cold medium. At different time intervals post infection (0, 6 and 12 h), intracellular virions were isolated on 4G2 antibodies, resolved by gel electrophoresis and detected by autoradiography as shown in the schematic (Fig. 4a). To assess and quantitate its intracellular distribution, we treated immunoprecipitated E with either endoglycosidase H (EndoH) or PNGaseF to measure its glycosylation status as read-outs for localisation within the secretory pathway. The viral E protein undergoes high mannose glycosylation in the ER, where it remains EndoH sensitive; upon arrival at the Golgi, it acquires complex glycans and becomes EndoH-resistant. In addition, the host protease furin cleaves prM to generate soluble pr and membrane-bound M, which can be resolved by gel electrophoresis (Fig. 4b). The post-ER pool of E protein was therefore calculated as the EndoH-resistant fraction of total intracellular E protein. In the Lyn-deficient cells or those expressing its mutant variants, the kinetics of ER-to-Golgi transport of E protein remained unaffected, as measured by acquisition of EndoH resistance (Fig. 4b, c). Secretion of virus particles is typically accompanied by appearance of membrane-bound E protein at the cell surface. To monitor this fraction, we first biotinylated the surface proteins from mock and infected cells, isolated them on streptavidin beads and detected by immunoblotting with anti-E antibodies (Fig. 4d). Contrary to ER–Golgi transport, arrival of E protein at the cell surface was impaired in $Lyn^{-/-}$ cells, or those expressing Lyn mutants (Fig. 4e).

To determine whether the block in secretion was specific to flavivirus particles or other cellular cargo too, we tested a range of different reporters as controls. First, we measured bulk cellular secretion of [$^{35}$S]cys/met-labelled proteins in supernatants of wt- and Lyn-deficient cells, which did not display any discernible difference (Fig. 4f). We also measured transport of transfected Vesicular Stomatitis Virus Glycoprotein and Influenza Hemag-glutinin (Fig. 4g–i). Neither of them displayed any transport defect in $Lyn^{-/-}$ cells. Secretion of Human coronavirus 229E, an unrelated +RNA coronavirus, also did not display any defect in $Lyn^{-/-}$ cells (Supplementary Fig. 4a, b). On the other hand, Dengue virus NS1 displayed ~50% reduction in secretion into supernatants in $Lyn^{-/-}$ cells, indicating that at least a fraction of NS1 is secreted through the Lyn-dependent pathway (Fig. 4j). Collectively, these data indicate that this pathway is specifically

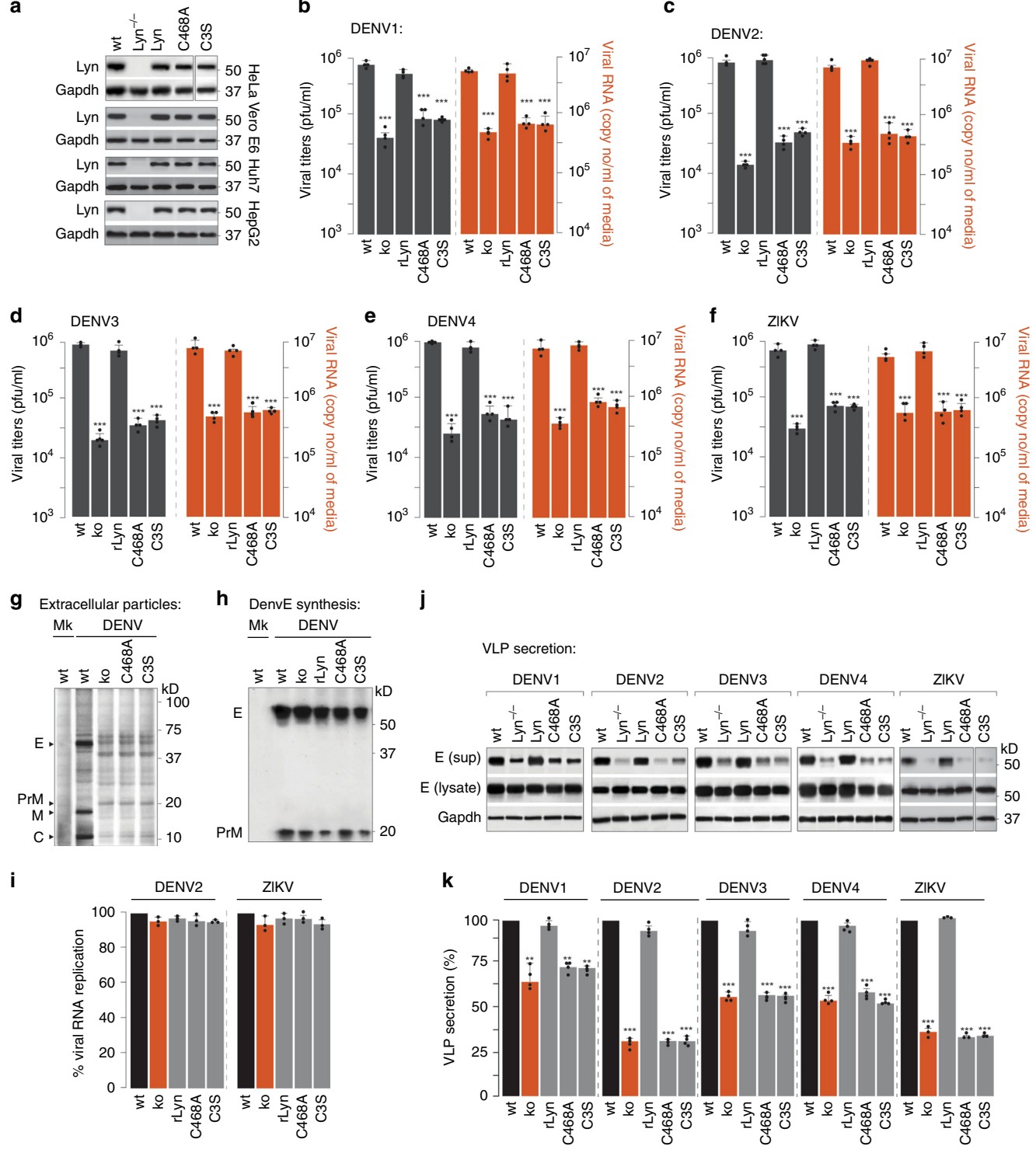

triggered upon flavivirus infection, where Lyn possibly phosphorylates transport proteins necessary for virus secretion.

To visualise the block in virus secretion, we imaged the distribution of viral E protein in wild-type and Lyn$^{-/-}$ cells, either infected with Dengue or Zika (Fig. 4k, l). In wild-type cells, E protein displayed its characteristic accumulation in membrane sites proximal to the *trans*-Golgi network, and also in LC3+ compartments. On the other hand, in Lyn-deficient cells, we detected aberrant E-protein distribution, which appeared to be in internal vesicles distinct from that of the *trans*-Golgi and colocalised largely with lysosomal compartments, which were

LC3+ at early time points, but became LC3− at later time points due to degradation. Collectively, our data indicate that in the absence of Lyn or in the presence of kinase-inactive or palmitoylation-deficient Lyn, synthesis, assembly and ER-to-Golgi transport of progeny virions remain unaffected; however, they are very likely missorted into vesicles that fail to be secreted, but fuse with lysosomes instead.

**Lyn is required for secretion of viral progenies in LC3+ secretory organelles.** SFKs are equipped with glycine and cysteine

**Fig. 3 Characterisation of Lyn activity in flavivirus secretion. a** Lyn deletions generated by two guide RNA (sgRNA) (5′-TGAAAGACAAGTCGTCCGGG-3′ and 5′-GTAGCCTTGTACCCCTATGA-3′) targeting human Lyn mRNA cloned into the chimeric CRISPR/Cas9 vector PX459. Transfected cells were selected on puromycin, and serially diluted to isolate single clones. Cells expanded from single clones were immunoblotted with anti-Lyn to verify deletion in the indicated cell lines. Lyn$^{-/-}$ cells were reconstituted with (i) wild-type Lyn, (ii) C468A kinase-inactive mutant or (iii) C3S palmitoylation-deficient mutant. Validation of gene deletion and expression of Lyn mutants were performed by immunoblotting. **b–f** Huh7 cells described in **a** were challenged with different serotypes of Dengue or Zika virus at MOI 2. Supernatants from infected cells were collected 48 h post infection and measured for viral titres using plaque assays (grey bars). Viral RNA in the extracellular medium was measured using RT qPCR (red bars). Data represent mean ± s.d. of four independent experiments. ***$P < 0.001$ (compared with wild type by ANOVA followed by one-sided Dunnett's test). **g** Supernatants collected from mock- or Dengue-infected [$^{35}$S]cys/met-labelled cells were concentrated and resolved by SDS-PAGE and detected by autoradiography. **h** Viral protein synthesis in wild-type, Lyn$^{-/-}$ cells and those expressing Lyn mutants was determined by pulse labelling with [$^{35}$S]cys/met for 10 min. Viral envelope prME was isolated from cell lysates on anti-E 4G2 antibodies and detected by autoradiography. **i** Virus replication was measured in Huh7 cells described in **a**, expressing viral replicons carrying a luciferase reporter. Data represent mean ± s.d. of three independent experiments. **j** Supernatants were collected from VLP-secreting cells that were either wild type or Lyn$^{-/-}$, or reconstituted with (i) wild-type Lyn, (ii) C468A kinase-inactive mutant or (iii) C3S palmitoylation-deficient mutant. Concentrated VLPs were resolved by SDS-PAGE and immunoblotted with 4G2 antibodies. **k** VLP secretion was calculated as the ratio of extracellular to total VLPs. Data are presented as a percentage of the control (wild-type) value (mean ± s.d. of four independent experiments). **$P < 0.01$; ***$P < 0.001$ (compared with control by ANOVA followed by one-sided Dunnett's test).

at their N terminus, which are typically acylated with myristoyl and palmitoyl chains, respectively. S-palmitoylation has been shown to dictate the spatial distribution and trafficking characteristics of several proteins, including SFKs[28–30]. Given the detrimental effect on virus transport with palmitoylation-deficient Lyn mutant C3S, we first determined how acylation affected virus secretion. We radiolabelled wild-type cells with [$^3$H]myristoyl or [$^3$H]palmitoyl CoA, and either mock-treated or infected cells with Dengue virus (Fig. 5a). Individual kinases were isolated on selective antibodies and their acylation detected by autoradiography. Fyn acquired myristoyl and palmitoyl chains in both mock and infected samples; Src did not undergo any acylation regardless of infection, whereas Lyn displayed palmitoylation in virus-infected cells.

To determine their subcellular distribution, we performed biochemical fractionations of organelles as shown in the schematic (Fig. 5b). Modified from a previous study[31], this strategy can effectively enrich for membranes of the secretory and endolysosomal compartments (Fig. 5c). In mock-infected cells, Src and Lyn co-migrated primarily with Golgi markers, whereas Fyn appeared in both the Golgi and PM fractions, in line with previous reports[29]. However, upon infection in wild-type cells, Src displayed a broad distribution over all membranes with modest enrichment in the ERGIC compartments. On the other hand, Lyn and Fyn migrated with the Transferrin receptor, Rab11 and LC3, representative of recycling endosomes/autophagosomes depending on cellular physiology[32–34] (Fig. 5d).

We also measured the subcellular distribution of mutant Lyn variants and viral E protein using this fractionation strategy to determine whether compartmentalisation was perturbed in Lyn-deficient cells. Both Lyn C468A and C3S mutants displayed aberrant distribution along with Dengue E, in lysosomal compartments (Fig. 5e). We hypothesised that loss of Lyn palmitoylation or kinase activity resulted in defective virus trafficking, which was sorted into lysosomes for degradation. To test this hypothesis, we blocked lysosomal degradation in wild-type and Lyn$^{-/-}$ cells by treating with chloroquine. MG132-treated cells were processed in parallel to block the proteasomal degradation pathway. We recovered significantly more viral E protein from Lyn$^{-/-}$ cells compared to wild-type upon blocking lysosomal degradation (Fig. 5f). Our data therefore suggest that activation of Lyn and its appropriate membrane targeting through palmitoylation maintains anterograde transport of progeny virus particles. Treating cells with 2-bromopalmitate (2-BP), a selective inhibitor of palmitoylation, resulted in significant loss of VLP secretion, corroborating these results (Supplementary Fig. 4c, d).

To confirm our biochemical data, we performed confocal imaging of the individual SFKs in Dengue- and Zika-infected cells (Fig. 5g–i). In line with the biochemical data, Lyn displayed the highest colocalisation with TfR+ and LC3+ compartments, Src appeared on ER and Golgi compartments and Fyn was distributed between PM and partially with Golgi and endosomal compartments. We also verified these results by immunoprecipitation on antibodies to identify their interactors in Dengue-infected cells, which reflected their subcellular distribution (Supplementary Fig. 4e, f).

**Secretion of proteolytically processed infectious viral progenies occurs via Lyn-regulated LC3+ compartments**. To determine how progeny virions/VLPs triggered Lyn-dependent export via secretory organelles, we determined whether the viral envelope prME alone could trigger their biogenesis. To do so, we generated stable cells secreting recombinant VLPs carrying a furin-resistant prM (H98A) and another carrying a second mutation (E E62K) that renders it furin-sensitive as reported previously[35] (Fig. 6a). We purified [$^{35}$S]-labelled VLPs from supernatants and verified that these mutations conferred furin resistance and sensitivity, respectively, as detected by autoradiography. Furin-sensitive VLPs secreted from wild-type cells displayed the anticipated pattern of cleaved pr and M fragments, whereas those carrying furin-resistant VLPs comprised uncleaved prM (Fig. 6b). To determine whether any differences existed between transport kinetics of the VLP variants, we collected them at different time intervals from cells pulsed with [$^{35}$S]cys/met to quantitate secretion of newly assembled VLPs (Fig. 6c). Densitometric analyses revealed that secretion of furin-resistant immature VLPs was significantly slower compared to processed, mature VLPs (Fig. 6d). Given their distinct transport kinetics, we further assessed whether this was Lyn-dependent. Interestingly, whereas wild-type and furin-sensitive VLPs followed Lyn-dependent secretion, the furin-resistant VLPs displayed no secretion defect in Lyn$^{-/-}$ cells, indicating that they followed a different transport mechanism (Fig. 6e). To measure their subcellular distribution, we separated homogenates of VLP-producing cells on sucrose and OptiPrep gradients as described in Fig. 5. Wild-type and furin-sensitive VLPs co-migrated with Rab11 and Tfr as seen with Dengue virions. However, furin-resistant VLPs were found enriched in the ER/ERGIC compartments instead (Fig. 6f). To isolate the distinct VLP-containing vesicles, we concentrated the gradient fractions enriched in wild-type, H98A and H98A/E62K VLPs and resolved them by gel electrophoresis (Fig. 6g). The lanes were then sliced into 2-mm sections and subjected to trypsin digest followed by mass spectrometry. Candidates identified in the wild-type and furin-sensitive VLP fractions

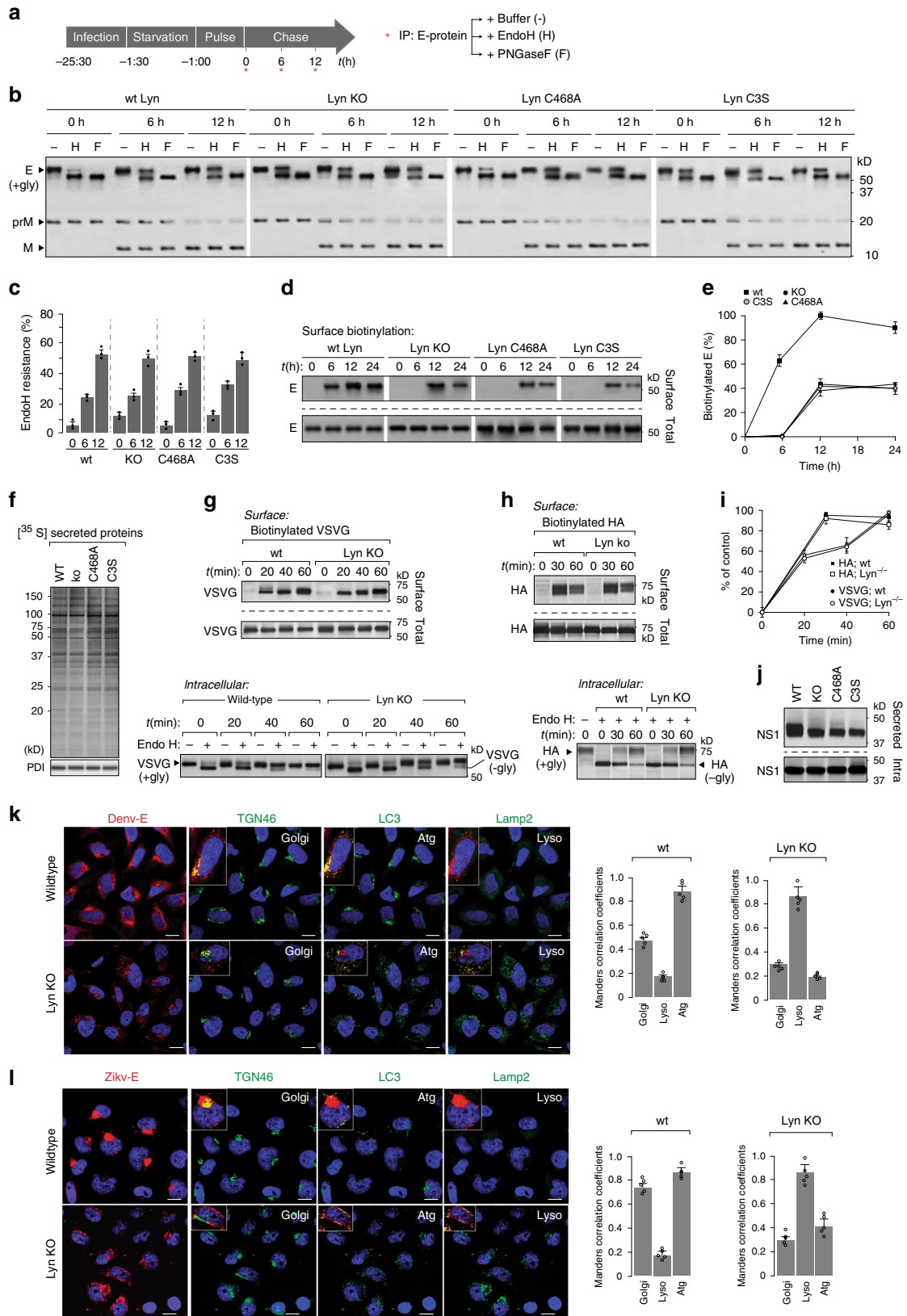

represented factors previously implicated in the secretory autophagy/amphisome machinery[36–39], whereas those isolated with the furin-resistant VLPs reflected vesicles of the conventional secretory pathway (Fig. 6h).

To test whether autophagy played a role in VLP secretion, we measured the appearance of LC3-I/II as autophagosomal markers, using rapamycin-induced autophagy as positive controls (Fig. 6i). To distinguish between secretory versus degradative autophagosomes, we generated control and VLP-secreting cells expressing a reporter for autophagic flux. RFP–GFP–LC3 cells appear as diffuse cytosolic fluorescence. Upon induction of autophagy, they appear as punctae—yellow in autophagosomes

**Fig. 4 Lyn deficiency blocks virus secretion post ER to Golgi transport. a** Schematic of pulse-chase analyses to characterise virus trafficking. **b** Wild-type and Lyn$^{-/-}$ Huh7 cells, or those reconstituted with C468A or C3S were infected with Dengue at MOI 2. Cells were radiolabelled and chased for 6 and 12 h. At each time point, virions were isolated and treated with (i) buffer only (−), (ii) endoglycosidase H or (iii) PNGaseF and detected by autoradiography. **c** Quantitation of EndoH resistance as % of total E. Error bars represent mean ± s.d. of three biological replicates. **d** Surface biotinylation on intact cells described in **b** was performed by Sulfo-NHS-biotin using the manufacturer's protocol. Lysates prepared from all cells were divided into two for (i) immunoprecipitation on streptavidin beads and (ii) total intracellular E expression. Biotin-modified E and total intracellular E from all samples were detected by immunoblotting with 4G2 antibodies. **e** Quantitation of biotinylated E was performed by densitometric analyses as the ratio of the surface to total E. Data are presented as a percentage of the control (wild type) (mean ± s.d. of three independent experiments). **f** Cells were labelled with [$^{35}$S]cys/met as described in **a**. Supernatants collected from wt or Lyn-deficient cells were resolved by SDS-PAGE and detected by autoradiography. **g, h** Wild-type and Lyn$^{-/-}$ Huh7 cells expressing either VSVG or influenza HA were pulse-labelled for 10 min and chased for the indicated time intervals. Surface fractions of VSVG or HA were biotinylated and immunoprecipitated (upper panel); intracellular pools were treated with EndoH to determine the distribution between the ER and Golgi. **i** Quantitation of biotinylated VSVG and HA was performed by densitometric analyses on immunoblots from **g, h**, and calculated as the ratio of the surface to total. Data are presented as percentage of the control (wild-type) value (mean ± s.d. of three independent experiments). **j** Secreted Dengue NS1 was measured in supernatants by immunoblotting. **k, l** Mock- and Dengue/Zika-infected cells were visualised by confocal imaging. Scale bar 20 μm. Colocalisation of prME with indicated organelle markers was quantified using Mander's correlation coefficient from 50 cells per condition (error bars represent mean ± s.e.m. of five independent experiments).

and eventually red in autolysosomes because of the acid-sensitive GFP signal. This reporter is therefore able to distinguish between non-degradative autophagosomes and autolysosomes. Using this reporter, we determined that control cells displayed basal levels of autophagosomes and autolysosomes (~10–20/cell for each), whereas VLP-secreting cells contained a significantly higher proportion of autophagosomes (~100/cell), and very few autolysosomes (5–10/cell). In contrast when treated with rapamycin, these cells had an abundance of autolysosomes (>80/cell) (Fig. 6j, k). These results could also be recapitulated in cells infected with either Dengue or Zika virus when cultured in fatty-acid-rich medium to bypass lipophagy (Supplementary Fig. 7a–d). These results support our mass spectrometry data that VLP secretion triggers formation of autophagosome-derived organelles that are non-degradative. Based on these data, we proposed a working model in which mature viral progenies trigger biogenesis of specialised secretory organelles derived from autophagosomes that facilitate efficient secretion of virus particles. These LC3+ organelles that are either secretory autophagosomes or amphisomes (generated upon fusion of the autophagosomes with multivesicular bodies or late endosomes) in the absence of Lyn, fuse with the lysosomal compartment for degradation. Immature virions on the other hand are unable to trigger this pathway and are instead exported through bulk exocytosis as illustrated (Fig. 6l).

**Infectious viral progenies are predominantly secreted via autophagosome-derived organelles.** To attempt to isolate viral progenies within membrane-bound compartments from the extracellular space, we collected the secreted population and separated them on a discontinuous sucrose gradient as shown (Fig. 7a). We collected 20 fractions and measured their infectivity using plaque assay and viral RNA amounts using RT qPCR. We observed two distinct peaks of viral particles—the lighter population (fraction II) was significantly more infectious compared to the denser population (fraction I) (Fig. 7b). To distinguish between free and membrane-bound virus populations, we measured their sensitivity to antibodies in the absence or presence of detergents. While the denser population remained sensitive, the lighter population reacted with antibodies only in the presence of 1% NP-40, suggesting that they remained protected within membranes (Fig. 7c). This was corroborated by biochemical analysis of the two populations. While both fractions (fraction I and II) were positive for viral E and prM, only the membrane-bound lighter fraction (fraction II) was positive for LC3 and Rab11 (Supplementary Fig. 7e). To characterise the released viral populations in Lyn$^{-/-}$ cells, we separated them on sucrose

gradients as described in Fig. 7a and determined that release of only the membrane-bound population was inhibited (Fig. 7d). To visualise whether viral progenies appeared within autophagosomes, we performed EM imaging on wt and Lyn$^{-/-}$ hepatocytes infected with Zika virus (MOI 2, 18 h). While in the wt cells virus particles appeared within double-membrane vesicles, Lyn$^{-/-}$ cells displayed an accumulation of these virion-containing organelles with a concominant increase in lysosomes (Fig. 7e). To characterise the infectivity of the two extracellular virus populations, we separated infectious venus-labelled Zika reporter virus[40] over sucrose gradients. The two separate populations were collected and used to infect hepatocytes. After 48 h, virus spread was measured by FACS (Fig. 7f). The lighter membrane-bound population (Frac II) displayed significantly higher spread compared to the denser population (Frac I), indicating that they might undergo cell-to-cell spread via distinct mechanisms.

To establish a functional link between secretory autophagosomes and virus secretion, we generated systematic depletions of components implicated in secretory autophagy, e.g., Grasp55/65[16,41], Trim16[42], SNARE complexes[42], Ulk1 complex and Rab GTPases[16] (Fig. 7g). Virus production was significantly attenuated by depletion of secretory autophagosomal components, but remained unaffected or moderately increased upon depletion of degradative autophagosomal components, e.g., Stx17 (Fig. 7g–k and Supplementary Fig. 6a–e). These results were also corroborated in primary human CD14+ monocytes (Supplementary Fig. 5). As reported for other cargo of the secretory autophagy pathway, we observed a more significant loss in infectious virus production from GRASP55-depleted cells compared to GRASP65[43,44]. Secretion of the VLPs in cells depleted in the Ulk1, SNARE and GTPase complexes also displayed ~60% reduction, reflecting the loss in the Lyn-dependent infectious population (Fig. 7l, m). On the other hand, depletion of the individual GRASP proteins did not affect VLP secretion, most likely due to compensatory effects. Collectively, our data indicate that secretory autophagosomes are generated in a virus-triggered Lyn-dependent manner, and play a significant role in secretion of mature infectious viral progenies. These results define an alternate route for flavivirus secretion, and indicate that at least a population of released viral particles is equipped to evade circulating antibodies for more efficient spread.

**Discussion**

In the current study, we investigated the role of Src-family kinases (SFKs) in flavivirus secretion. SFKs were recently described to play a KDELR-dependent signalling role at the Golgi by triggering anterograde protein transport during increased flux through

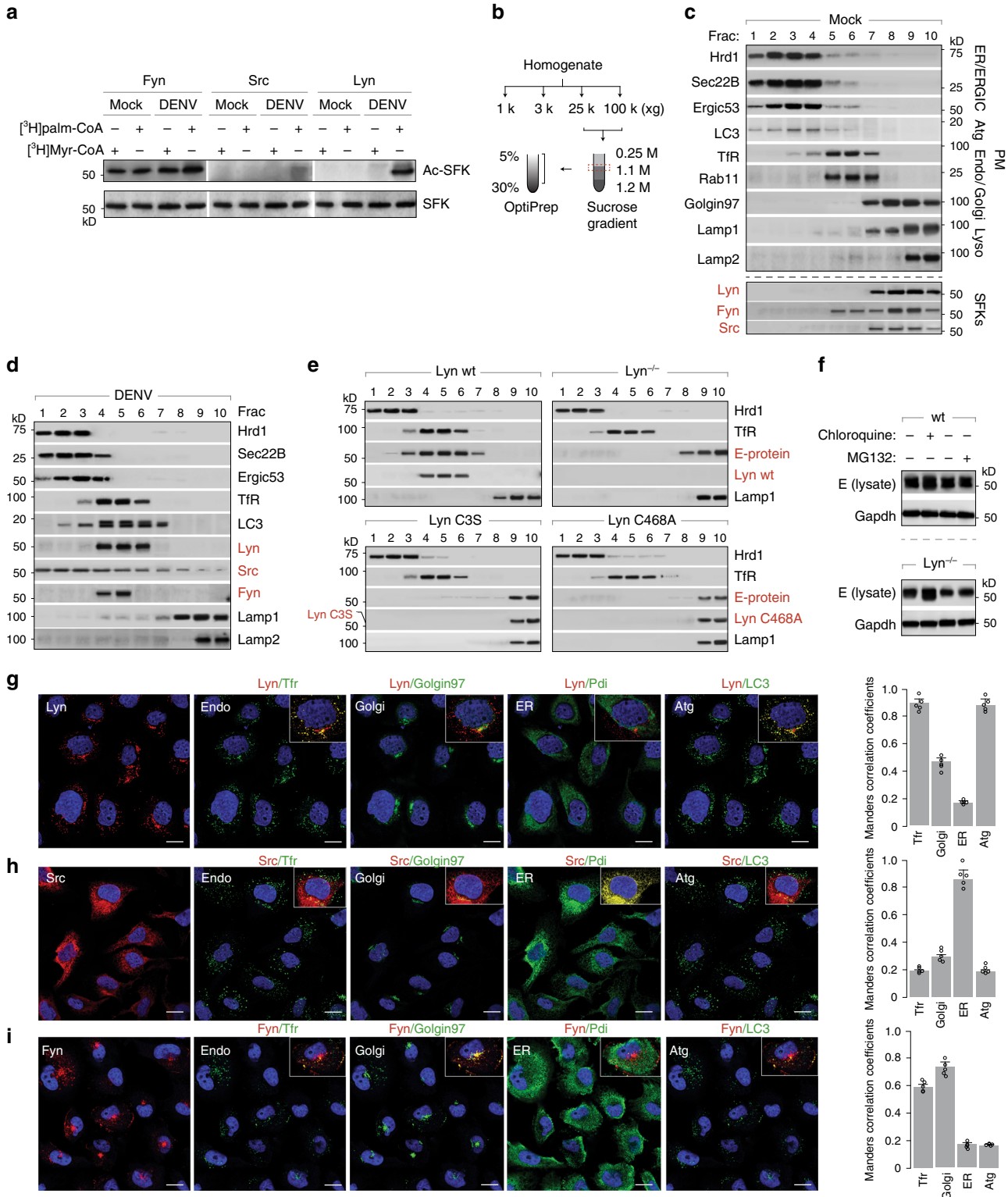

the secretory pathway[24]. Since Dengue virus is known to hijack the KDELRs for trafficking to the Golgi, we hypothesised that a similar signalling cascade might be activated that facilitated virus secretion. Furthermore, SFK inhibitors, such as dasatinib and AZD0530, were previously reported to efficiently block Dengue infection by affecting virus assembly and secretion[45].

To test our hypothesis, we screened SFKs that were activated during virus infection. We identified three members—Src, Fyn and Lyn—that displayed increased phosphorylation in their

activation loop in cells infected with Dengue or Zika. This phenomenon was recapitulated in cells that constitutively secreted the corresponding VLPs. Pharmacological and genetic inhibition of the SFKs, Lyn in particular, significantly attenuated secretion, both in infected and VLP-producing cells. Although the basic secretory machinery is conserved in all eukaryotic cells, it is likely that virion transport and release may proceed differently with contributions from multiple SFKs, depending on specific cell types, exemplified by partial rescue of secretion in Lyn-deficient

**Fig. 5 Virus secretion requires palmitoylation and kinase activity of Lyn. a** Wild-type Huh7 cells were radiolabelled with [³H]myristoyl or [³H]palmitoyl CoA and either mock- or Dengue-infected (MOI 2, 48 h). Lyn, Fyn and Src were immunoprecipitated, resolved by SDS-PAGE and detected by autoradiography. Total expression levels were measured by immunoblotting in cell lysates. **b** Schematic for biochemical fractionation to isolate intracellular compartments of the secretory/endolysosomal compartments. **c** Separation of organelle markers to verify membrane fractionation using the schematic in **b**, from mock-infected cells. **d** Intracellular distribution of organelles and the three SFKs was measured in wild-type Huh7 cells upon Dengue virus infection (MOI 2, 48 h) using the biochemical separation strategy described in **b**. **e**. Intracellular distribution of progeny virions (E protein) and Lyn variants was measured in wild-type and Lyn⁻/⁻ cells or those expressing C468A or C3S Lyn mutants, using the biochemical separation strategy described in **b**. **f** Wild-type and Lyn⁻/⁻ cells were Dengue-infected (MOI 2, 48 h) and (i) untreated, (ii) chloroquine treated (100 μM) or (iii) MG132 treated (40 μM). Expression of viral E protein was measured by immunoblotting with 4G2 antibodies **g**–**i**. Confocal imaging of Lyn, Src and Fyn distribution in Dengue-infected cells (MOI 2, 48 h). Scale bar 20 μm. Mander's correlation coefficients were calculated for colocalisation between individual SFKs and organelle markers for endosomes, Golgi, ER and autophagosomes from 50 cells per condition (error bars represent mean ± s.e.m. of five independent experiments).

cells reconstituted with Lck (Supplementary Note 1 and Supplementary Fig. 8).

We first confirmed that attenuated virus production in Lyn-deficient cells was specifically due to a block in secretion and not entry or replication. In cells deficient in Lyn, or expressing kinase-inactive or palmitoylation-deficient mutants, entry and replication remained unaffected, as did kinetics of transport from the ER to the Golgi compartments. On the other hand, post-Golgi transport was disrupted, resulting in a significant loss of secreted viral progenies/VLPs. This defect could be rescued when Lyn⁻/⁻ cells were reconstituted with wild-type Lyn, but not an enzymatically inactive or palmitoylation-deficient mutant.

To further characterise the specific block in virus transport sustained in Lyn-deficient cells, we biochemically separated organelles of the secretory/endolysosomal pathway from infected cells. In wild-type cells, both Lyn and virions were enriched in membranes that were positive for Rab11, the transferrin receptor and LC3. These markers have been reported to be enriched in both recycling endosomes, as well as exosomal and secretory autophagosomal vesicles. Contrary to wild-type Lyn, upon expression of the kinase or palmitoylation mutants, virions partitioned with lysosomal fractions. Inhibiting lysosomal degradation could rescue a significant fraction of viral E protein. Interestingly, the mode of Lyn-dependent transport was specifically triggered by processed, mature virions; conventional secretion of immature virions and cellular cargo occurred in a Lyn-independent manner. Isolation of the different populations of extracellular virus particles revealed that the Lyn-dependent population was membrane-bound, infectious, resistant to antibodies and were able to spread more rapidly compared to the free particles. Systematic depletion of genes implicated in secretory autophagy supported the hypothesis that virus release depended on this pathway.

Current evidence on the mechanisms of flavivirus secretion is limited, especially at the post-Golgi steps. Class-II ADP-ribosylation factors and KDELRs have been described for Dengue transport at a pre-Golgi step[46]. Although several components of the endosomal sorting complex (ESCRTs) were identified in Dengue and Japanese Encephalitis virus infection, they were found to participate in budding of immature virus particles into the ER lumen rather than at the later stages of secretion[47]. More interesting observations have emerged from Zika infection, where EM studies have identified large vesicles containing multiple virions, as well as small vesicles containing individual virions en route to fusion at the plasma membrane[48,49]. Ultrastructural studies on the morphogenesis of Zika virus in mammalian cells have also revealed infrequent single virions in the Golgi and virus clusters within convoluted membranes, believed to emerge from the ER and ERGIC compartments[50,51]. Groups of virions were observed to be released upon fusion of virus-containing vacuole-like organelles with the plasma membrane[48]. Our data are in agreement with multiple exit routes being exploited by these viruses; however, egress as virus clusters within membranes very likely results in more infectious virus populations.

The proviral effect of autophagy in flavivirus infections has been documented for several members such as Dengue[52], Zika[53], HCV[54] and Japanese Encephalitis[55]. One exception is West Nile Virus, where autophagy does not appear to play any significant role[56]. Arbovirus infections in mosquito cells have also suggested a limited role of this pathway in the viral life cycle[57]. We previously reported on the induction of selective autophagy for lipid droplet hydrolysis[58]. Our current results suggest that besides lipophagy, viral progenies also exploit autophagosome-derived secretory organelles to exit host cells, underscoring the importance of autophagy in multiple stages of the viral life cycle. This study provides insights into modes of virus secretion with important implications in mechanisms of cell-to-cell transmission, spread and tissue tropism of flaviviruses.

## Methods

**Cells lines, viruses and antibodies**. The following cell lines were used in this study: BHK21, Vero E6, HeLa, Huh7 and HepG2 cells obtained from ATCC. Cells were maintained in DMEM or EMEM supplemented with 10% foetal bovine serum (FBS) and 1% penicillin/streptomycin at 37 °C with 5% CO₂. The stable cell lines expressing prME-DENV1–4, prME-ZIKV or Lyn WT/C468A/C3S mutants were established using the retroviral vector pCHMWS-IRES-Hygromycin, selected following a 2-week period in the presence of 500 μg/mL hygromycin and maintained thereafter in the same medium[8,22]. For biochemical analyses, the following antibodies were used: mouse anti-E mAb 4G2 (dilution 1:1000) from Novus Biologicals, or prepared using hybridoma cells D1-4G2-4-15 from ATCC; rabbit anti-phosphorylated Src family mAb (1:1200), rabbit anti-Lyn mAb (1:1500), rabbit anti-Src mAb (1: 1000) and rabbit anti-Fyn (1:1000) from Cell Signaling Technology; mouse anti-Gapdh mAb (1:2000) from Abcam. Phospho-Tyrosine Mouse mAb (P-Tyr-100)-conjugated magnetic bead from Cell Signaling Technology (dilution 1:100).

Virus stocks were prepared for DENV2 (16681), DENV1 (Hawaii), DENV2 (New Guinea), DENV3 (H87), DENV4 (Jamaique 8343) and ZIKV (MR766) by determining tissue culture-infective dose 50% (TCID₅₀/ml) in Vero E6 cells challenged with 10-fold serial dilutions of infectious supernatants for 90 min at 37 °C. Cells were subsequently incubated in DMEM with 2.5% FCS. HCoV-229E (ATCC VR-740) titres were determined by TCID₅₀/ml in Vero E6 cells.

**Virus infections**. *RT-qPCR assay to measure virus infection*: Cells were plated in 96-well plates and infected at a MOI of 2. Infected cells were collected at indicated time intervals (as specified in the figure legends). For quantitation of RNA, cells were washed first with PBS and collected in 250 μl of Trizol reagent for isolation of total RNA. Real-time PCRs were performed with one- or two-step methods using SYBR green or Taqman chemistry with gene-specific primers provided in Supplementary Table 1.

*Plaque assays*: Serial dilutions of supernatants from infected cells were performed by adding on to BHK21/Vero E6 monolayers. After adsorption for 60 min at 37 °C, cells were washed and plaque media was overlaid on the cells. After 3–6 days of incubation at 37 °C, the monolayers were stained with crystal violet and plaques were counted.

*Replicon assay*: Dengue and Zika *Renilla* luciferase replicons were transcribed using a Ribomax T7 RNA polymerase kit (Promega, Madison, WI). The resulting RNA was purified by sodium acetate ethanol precipitation. Cells were washed with PBS, resuspended in electroporation buffer (Teknova), followed by transfection with *Renilla* luciferase replicons and harvested at various times post transfection as indicated. Luciferase expression was measured using Renilla Luciferase Assay

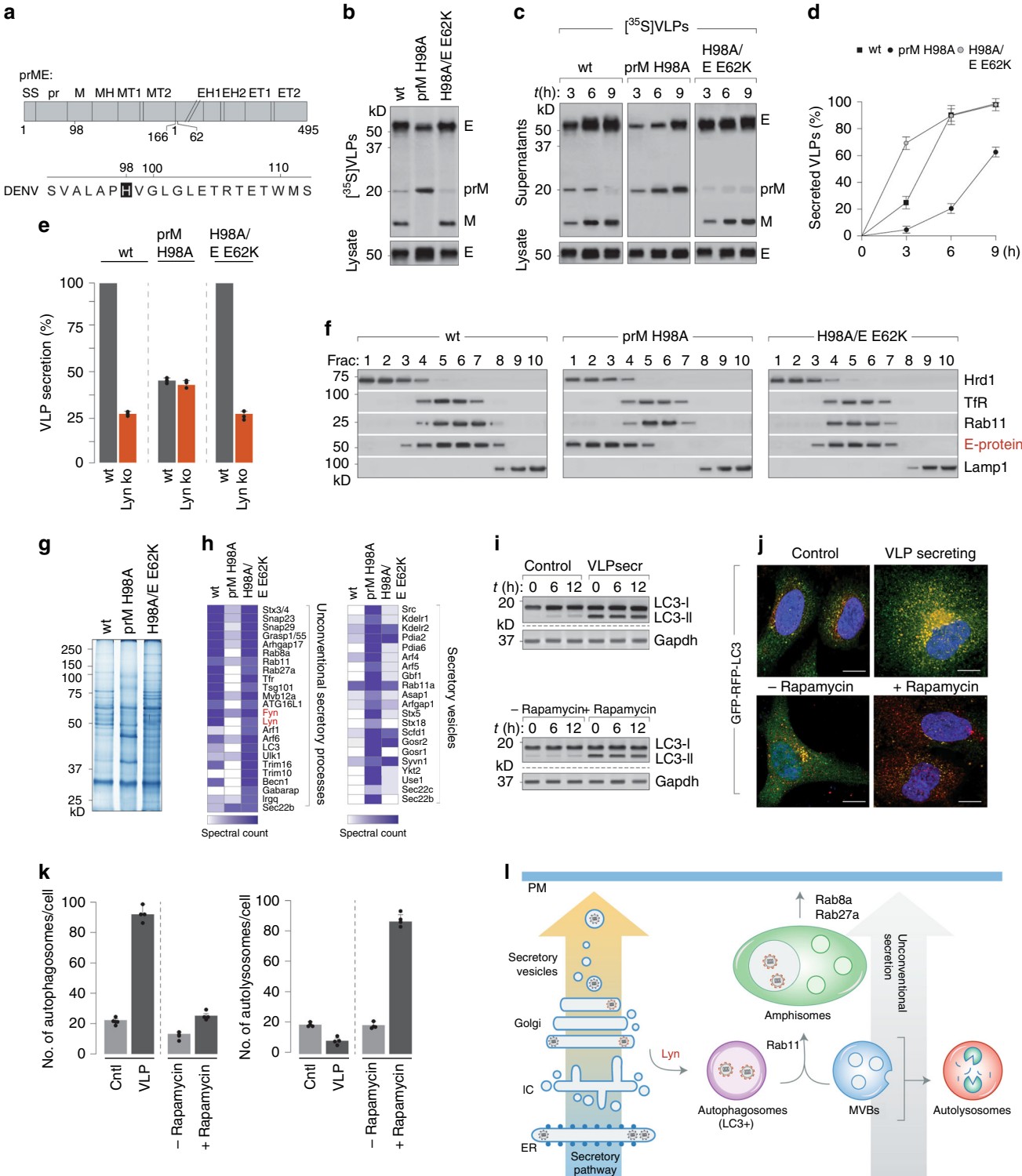

system (Promega). A replication-defective NS5-GDDm mutant was electroporated in parallel as negative control, where indicated.

**Mass spectrometry**. To screen phosphorylated SFKs in Dengue infection, $3 \times 10^6$ Vero E6 or BHK21 cells were seeded a day before infection. Attached cells were challenged by DENV1 with a MOI of 2 and harvested at 1 day post infection. Harvested cells were lysed on ice with 1 ml of RIPA buffer (1% Triton X-100, 150 mM NaCl, 50 mM Tris-HCl (pH 7.5), 1 mM EDTA and 0.5% Na-deoxycholate) supplemented with freshly added protease inhibitor cocktail (Roche) and phosphatase inhibitor tablets (Roche) for 30 min. Immunoprecipitation was performed on anti-phospho-SFK antibodies (1:500) conjugated to magnetic beads using a magnetic conjugation kit (Abcam, ab269890). Eluates were separated by gel

electrophoresis and visualised by sliver staining. The entire lanes were sliced into 2-mm sections and were further processed in 50% water/methanol as previously described[59,60]. Samples were trypsinised and subjected to an LTQ Orbitrap mass spectrometer for identification of candidates. To identify host proteins co-migrating with the VLPs (Fig. 6), VLP-enriched fractions were concentrated, resolved by SDS-PAGE and processed as described above. MS/MS spectra were analysed using Sequest algorithm searching a composite target-decoy protein sequence database. The target sequences comprised the human protein repository of the Uniprot database. Decoy sequences were obtained upon reversing the orientation of target sequences. Allowed criteria for searches required trypsin cleavage (two missed cleavages allowed), peptide mass tolerance of 20 p.p.m, variable oxidation of methionine residues and static carbamylation modification of cysteine residues. Peptide-spectrum matches were determined with estimated

**Fig. 6 Lyn-dependent virus transport from the Golgi is specifically triggered by processed, mature virions. a** Schematic of prME with mutations that confer furin resistance. **b** Radiolabelled VLPs, either wild type or carrying indicated mutations, were purified from supernatants of [$^{35}$S]cys/met-labelled cells. VLPs were resolved by SDS-PAGE and detected by autoradiography. **c** Time courses of VLP secretion were measured from radiolabelled wild-type cells and detected by autoradiography. **d** Quantitation of VLP secretion was performed by densitometric analyses on the autoradiogram from **c** to measure the fraction of secreted-to-total E protein, and presented as % of control (wild type, maximum secretion set at 100%). Error bars represent mean ± s.d., *n* = 3. **e** VLP secretion for wild-type, H98A and H98A/E62K variants was measured from wild-type (grey bars) and Lyn$^{-/-}$ cells (red bars) as described in **c**, quantitated by densitometric analyses and presented as % of control (wt VLPs secreted from wt cells, set at 100%). Error bars represent mean ± s.d.; *n* = 3. **f** VLPs (wild type, H98A and H98A/E62K) were fractionated on sucrose and Optiprep gradients as described in Fig. 5 to detect their subcellular distribution. **g** VLP-containing fractions co-migrating with Lyn on OptiPrep gradients described in **f** were scaled up, resolved by gel electrophoresis and detected by Coomassie staining. **h** Whole lanes were sliced into 1-mm sections and subjected to trypsin digest. The peptide mix was processed and analysed by LTQ Orbitrap mass spectrometer. **i** Control and VLP-secreting cells were immunoblotted for LC3-I/II; rapamycin-induced autophagosomes served as controls. **j** Control or VLP-secreting cells (upper panel) were stably transfected with GFP–RFP–LC3 reporter; control cells cultured in media ±rapamycin (lower panel). Scale bars, 10 μm. **k** Formation of autophagosomes was measured by counting fluorescent puncta that were GFP+ and RFP+ (*n* =~1000 cells) and presented as abundance per cell. Autolysosomes were measured by quantitating RFP+red puncta over ~1000 cells. Error bars represent mean ± s.d.; *n* = 4. **l** Schematic of the secretion for immature versus processed mature virions through the conventional secretory pathway versus unconventional secretory organelles.

---

false-discovery rate <1%. Spectral counts for each condition were combined at a protein level and normalised by protein length to infer protein abundances and intensities in each case. The criterion for selecting candidates from the mass spectrometry dataset was identification of at least two unique peptides. Proteins presented in the panel were those that were considered statistically significant after imposing a criterion of Log$_2$[fold enrichment]>4 between control and sample set. Identified hits were further categorised into different biological pathways using Panther and Ingenuity Pathway Analyses software.

**Immunoprecipitation (IP) assay**. Cell lysates (CL) were pre-cleared by incubation with 30 μl of 50% Protein G Sepharose beads (Amersham Pharmacia Biotech) for 1 h. Pre-cleared lysates from mock- or DENV-infected cells were then incubated for overnight at 4 °C with 30 μl of 50% Protein G Sepharose beads conjugated to pSFKs (for phosphorylated SFK screen) or P-Tyr-100-conjugated magnetic beads (to validate selected SFK candidates). Subsequently, beads were collected by centrifugation at 13,000 rpm for 30 s at 4 °C and washed three times with cold wash buffer (50 mM Tris-HCl buffer, pH 7.4, containing 0.1% (wt/vol) Triton X-100, 300 mM NaCl and 5 mM EDTA) supplemented with 0.02% (wt/vol) sodium azide and phosphatase inhibitor tablets, and once with cold PBS. Bound proteins were eluted by boiling in 30 μl of 2×SDS-PAGE loading buffer, separated by gel electrophoresis and visible in silver stain or analysed by western blotting using appropriate antibodies.

**Luminex assay**. MILLIPLEX MAP 8-Plex Human SFK Phosphoprotein Kit was purchased from Millipore and used according to the manufacturer's instructions. Gapdh MAPmate kit was used as loading control. Cell lysates from virus-infected cells were collected and protein concentrations adjusted to 1 mg/ml. In total, 25 μl of each lysate preparation was mixed with 8-Plex SFK bead set and Gapdh beads per well of a 96-well plate. Following incubation overnight at 4 °C on a slow shaker, 25 μl of 1× biotinylated antibody mixture was added to each well. After incubating for 1 h at room temperature with slow shaking, 25 μl of 1× streptavidin–phycoerythrin solution was added to each well. After incubating for 15 min at room temperature, 25 μl of amplification buffer was added to each well and incubated for 15 min at room temperature. The reaction mix was resuspended in 150 μl of assay buffer and analysed on Luminex 200$^{TM}$ analyser. Median fluorescence intensities (MFI) for triplicate wells were averaged and normalised to average MFI of Gapdh as loading control.

**Lentivirus transduction of cells**. For production of lentivirus stocks, sub-confluent 293T cells were transfected with packaging plasmids pCMV-Gag-Pol, pMD2VSV-G and specific plasmids. Two days post transfection, lentivirus-containing medium was collected, filtered and titrated. Stable cell lines were produced by transduction of the target cells with lentivirus particles at an MOI of 0.1 followed by selection with puromycin-containing media[61].

**siRNA experiments**. All siRNAs used in this work, including non-targeting (NT) siRNA (D-001206) and transfection reagent DharmaFECT1 (T-2001), were purchased from Dharmacon and provided in Supplementary Table 3. Src siRNA (L-003175), Fyn siRNA (L-003140) and Lyn siRNA (L-003153) were provided as SMARTpool ON-TARGET plus siRNAs, which are pools of four siRNAs targeting various sites in a single gene. For siRNA experiments, reverse transfection was performed using DharmaFECT1 reagents as recommended by the manufacturer. Briefly, siRNAs mixed with DharmaFECT1 reagents were added to 24-well plates in DMEM medium without FBS and antibiotics. Twenty minutes later, 0.8 ml cells (60,000 cells/ml in DMEM supplemented with 10% FBS) were added to each well to the final siRNA concentrations. Cells were then incubated at 37 °C for 72 h. For

VLP assays, the medium was replaced with 0.3 ml of Opti-MEM and, 14 h later, culture supernatant (SN) containing secreted VLPs was collected and clarified by centrifugation at 4000 rpm for 5 min. Cells separated from supernatants were lysed in RIPA buffer containing freshly added protease inhibitor cocktail (Roche) for 30 min on ice. For DENV infection assay, siRNA treatment was performed 48 h before viral infection using the same conditions described above, and the siRNA-transfected cells were re-seeded to avoid over-confluency before virus challenge. Cells and supernatants containing progeny viruses were collected and processed as described above.

**VLP quantification**. To detect VLP secretion, 90 μl of supernatants and cell lysates from prME expressing stable cell lines or transiently transfected cells were added with 30 μl of 4× NuPAGE LDS sample buffer and subjected to western blotting using anti-E antibody 4G2. The mean luminescence and area of E- protein signals detected were measured by densitometry using image J software.

**SU6656 treatment**. Src-family Kinase inhibitor—SU6656 was purchased from Selleckchem and dissolved in DMSO to make a 50 mM stock. Cell cytoxicity was first measured by MTT assay. To test VLP secretion in SU6656-treated cells, 2 × 10$^5$ DENV1–4 prME-expressing cells were pre-seeded a day before treatment. SU6656 with indicated concentrations as specified in the figures was added to cells. Control cells (0 μM) received the same amount of DMSO. After 6 h of treatment, the medium was changed into 500 μl of Opti-MEM supplemented with SU6656 or DMSO. Supernatants and cell lysates were collected 16 h later to measure secreted versus total VLPs, respectively.

**CRISPR/Cas9-mediated deletion of Lyn**. Potential target sequences for CRISPR interference were found using established rules[62]. SgRNA targeting of Lyn was designed (Supplementary Table 2) and cloned into the chimeric CRISPR/Cas9 vector PX459[8,59,60]. Upon confirming potential off-target effects of the seed sequence using NCBI human nucleotide blast, none of the bases overlapping with any other location of the human genome was found. The PX459-sgRNA clone was used to transfect HeLa, Vero, HepG2 and Huh7 cells (for infection assay). Cells were selected on puromycin, followed by limiting dilution to isolate individual colonies, which were expanded and maintained in culture. Deletion was verified by immunoblotting with anti-Lyn to verify deletion.

**Fluorescence microscopy**. For fluorescence microscopy, cells grown on coverslips were fixed in 4% paraformaldehyde for 15 min, permeabilised with 0.1% TX-100 in PBS for 5 min and blocked with 10% BSA, 5% goat serum and 50 mM glycine for 30 min. Cells were incubated with primary antibodies for 2 h at room temperature, and then probed with appropriate secondary antibodies. Nuclei were stained with DAPI and mounted on glass slides for image acquisition using either a Zeiss Observer Z.1 or an LSM 700 confocal microscope. To quantify colocalisations, images were analysed for colocalisation using Imaris 9.2, where a background subtraction was performed and a threshold for pixel intensity was automatically determined by the software. The Manders values shown indicate the overlap in the automatically determined region of interest.

**Transmission electron microscopy**. Wild-type and Lyn$^{-/-}$ Huh7 cells were infected with Zika (MOI 2, 18 h), fixed in 2.5% glutaraldehyde, washed three times in PBS and serially dehydrated. The cells were postfixed in 1% osmium tetroxide and embedded in Araldite resin (Polysciences, Inc., Warrington, PA). Blocks were sectioned with a diamond knife on an ultramicrotome (Leica microsystems) and examined with a transmission electron microscope (CM100, Philips).

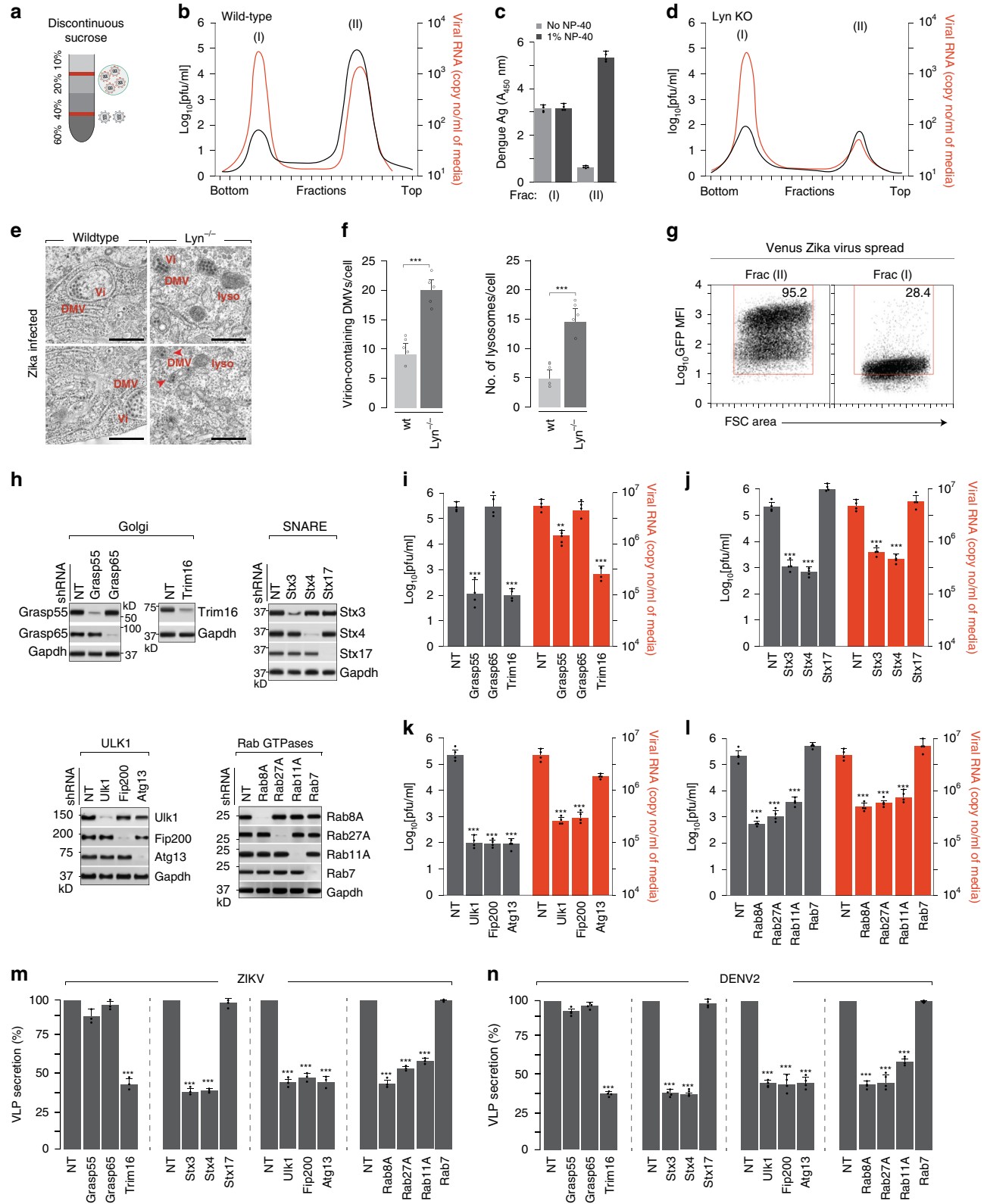

**Pulse-chase analyses of viral protein transport**. Pulse-chase experiments were performed as previously described[61,63]. Briefly, ~1 × 10^7 cells (mock or virus-infected) were detached by trypsinisation and starved for 30 min in cysteine/methionine-free medium at 37 °C prior to pulse labelling. Cells were labelled in 10 mCi/ml of [35S]Cys/Met for 10 min in a 37 °C water bath, and chased in cold medium for the indicated time intervals. At each time point, aliquots were withdrawn, and the reaction stopped with cold PBS. Cell pellets were either stored for further processing or labelled with NHS biotin for surface labelling. For measuring transport characteristics of E protein, cell pellets were lysed in Tris buffer

containing 0.5% NP-40. Lysates were pre-cleared with agarose beads for 1 h at 4 °C, followed by immunoprecipitations for 3 h at 4 °C with end-over-end rotation. Immunoprecipitated samples were eluted by boiling in reducing sample buffer and subjected to SDS-PAGE followed by detection using autoradiography.

**Biochemical fractionations for membrane separation of the secretory pathway**. Mock- and virus-infected cells were pelleted and washed three times with PBS. Cells were homogenised in 10 mM Tris buffer (pH 7.2) containing 400 mM

**Fig. 7 Secretion and spread of viral progenies is facilitated by secretory autophagosomes. a** Schematic for separation of infectious Zika virus populations over sucrose-step gradients. **b** Infectivity and vRNA in fractions collected from the gradient were measured by viral titration and RT qPCR. Data are presented as $\log_{10}$[pfu/ml] and Zika RNA copies/ml. Peak I; heavier fraction, Peak II; lighter fraction. **c** Zika E protein was detected by ELISA in the two peak fractions upon treatment ±1% NP-40. Error bars represent mean ± s.d., $n = 4$. **d** Supernatants collected from Lyn$^{-/-}$ cells were separated on sucrose gradients as described in **a**, and measured by plaque assay and RT qPCR. **e** Wild-type and Lyn$^{-/-}$ Huh7 cells were infected with Zika at MOI 2 for 18 h, and subcellular structures were visualised by transmission electron microscopy. Scale bars, 200 nm. **f** DMVs and lysosomes were quantified on blinded images; $n = 20$ cells/condition. Data are represented as mean ± s.e.m. of five biological replicates; ***$P < 0.001$ (compared with wild type by ANOVA followed by one-sided Dunnett's test). **g** Extracellular populations of venus-labelled Zika virus MR766 strain were separated over sucrose gradient described in **a**. Huh7 cells were infected with Fractions I and II, and spread of venus Zika was measured by FACS. Box indicates gating strategy on GFP+cells. **h**. Depletion of indicated proteins was performed by stable shRNA transductions in Huh7 cells, and verified by immunoblotting. **i**.–**l**. Cells described in **h** were cultured in medium supplemented with 10 mM BSA-conjugated fatty acids and infected with Zika at MOI 2 for 48 h, and measured for infectivity using plaque assays (grey bars) and extracellular vRNA using RT qPCR (red bars). Data represent mean ± s.d. of four independent experiments. **$P < 0.01$; ***$P < 0.001$ (compared with wild type by ANOVA followed by one-sided Dunnett's test). **m**, **n** Secretion of either Zika (left panel) or Dengue (right panel) VLPs in shRNA-depleted cells described in **g**. VLP secretion was calculated as thev ratio of extracellular to total VLPs. Data are presented as a percentage of the control (wild-type) value (mean ± s.d. of three (Zika) and four (Dengue) independent experiments); ***$P < 0.001$ (compared with control by ANOVA followed by one-sided Dunnett's test).

sucrose, 1 mM EDTA supplemented with protease inhibitor cocktail, using a loose pestle Dounce homogeniser. Homogenates were centrifuged at $100,000 \times g$ for 45 min to collect total membranes or centrifuged sequentially at $1000 \times g$ (10 min), $3000 \times g$ (10 min), $25,000 \times g$ (20 min) and $100,000 \times g$ (30 min, Beckman TLA100.3 rotor) to collect membrane fractions. The $25,000 \times g$ and $100,000 \times g$ fractions were combined and resuspended in 750 µl of 1.25 M sucrose buffer and overlaid with 500 µl of 1.1 M and 500 µl of 0.25 M sucrose buffer. Centrifugation was performed at $120,000 \times g$ for 2 h (Beckman TLS 55 rotor), after which the interface between 0.25 M and 1.1 M sucrose was collected and resuspended in 1 ml of 19% OptiPrep for a step gradient containing 0.5 ml 22.5%, 1 ml 19% (sample), 0.9 ml 16%, 0.9 ml 12%, 1 ml 8%, 0.5 ml 5% and 0.2 ml 0% OptiPrep each. The gradient was centrifuged at $150,000 \times g$ for 3 h (Beckman SW 55 Ti rotor), and subsequently ten fractions, 0.5 ml each, were collected from the top.

**Statistics and reproducibility**. Statistical analyses were performed with GraphPad Prism software. In all figures, the datapoints and bar graphs represent the mean of independent biological replicates. In all graphs, the error bars represent the standard deviation and are only shown for experiments with $n = 3$ or greater as indicated. For microscopy experiments, data sets for quantitative analysis were acquired from an average of 40–50 fields from four to five independent reproducible experiments for each condition. Comparisons between control and sample datapoints were made using either the Student's unpaired $t$ test with a confidence limit for significance set at 0.05 or less, or one-way ANOVA with Dunnett's multiple-comparison analyses post test, as specified in the figure legends.

**Reporting summary**. Further information on research design is available in the Nature Research Reporting Summary linked to this article.

## Data availability

UniprotKB accession codes of proteins identified by mass spectrometry are provided in Supplementary Data 1 and were extracted from UniprotKB (Human, release 2020_06 including isoforms and unreviewed sequences, https://www.uniprot.org/uniprot/). Protein sequences of Dengue and Zika strains were extracted from UniprotKB. All accession codes of RNAi experiments have been provided in Supplementary Tables 3 and 4. The remainder of the data generated or analysed during the current study are available from the corresponding author on reasonable request. Source data are provided with this paper.

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

## Acknowledgements

The authors acknowledge Nicla Porciello for help with Lck plasmids. This work was supported by Health and Medical Research Funds (16150592 and 16150732), Research Grants Council (17113019), the Scientific Research Plan of the Beijing Municipal Education Committee (KM201710025002) and the Croucher Foundation (to S.S.).

## Author contributions

M.Y.L. and T.S.N. carried out the majority of the experiments, interpreted the results and helped write the paper. L.Y.L.S., E.S., P.W., X.Y. and Y.L. conceived, carried out experiments and interpreted the results. R.B. and J.A. contributed to the concept of the study, O.A. and M.E. generated and contributed critical reagents for the study. S.S. contributed to the concept of the study, interpretation of the results, wrote the paper and supervised the project.

## Competing interests

The authors declare no competing interests.
