## [Peer Review File · Nature Communications]

Reviewers' Comments:

Reviewer #1:

Remarks to the Author:

In this revised manuscript that I had earlier reviewed for another Nature journal, Li et al. have very well addressed my major and minor comments. The authors conducted a comprehensive set of additional experiments, the results supporting the authors' conclusions. For instance, they now corroborate the link between virus secretion and post-Golgi autophagosomes, show that Lyn knock-down does not cause a general secretion defect and fill several gaps. I have no further comments and consider this an important contribution to the field and a well conducted set of experiments.

Reviewer #2:

Remarks to the Author:

In the present manuscript li et al. investigated the mechanism of flaviruses exit from infected cells. The previous version of the manuscript appeared somewhat preliminary because some molecular details and some critical experiments were missing.

In the revised manuscript the authors performed a number of key experiments to address the criticisms and modified the text and figures appropriately. The current version of the manuscript is much more robust and the experiments definitely support the conclusions.

In my opinion the manuscript is now suitable for publication in Nature Communications

Reviewer #4:

Remarks to the Author:

This is a much improved version of the manuscript previously submitted to Nat Microbiol.

The data about the involvement of Lyn in the secretion of flavivirus are very solid, and experiments are very well performed and controlled.

The involvement of secretory autophagosomes, however, is more circumstantial and the authors do not prove this point directly. Therefore, the conclusions regarding how flavivirus are secreted have to be played down and be more balanced.

Major criticisms:

Figures 6i-6k. These experiments have also to be repeated with DENV and ZIKA.

Figures 6j/6k. At this point of the manuscript, the authors have no prove that the yellow puncta are autophagosomes (even later). LAPosomes (or any other cytoplasmic LC3-positive structure) are also yellow when labelled with RFP-GFP-LC3 and as a matter of fact, structures recruiting non-lipidated LC3 appear as yellow puncta as well. Overall, the authors cannot conclude that the yellow puncta that they are detecting at are secretory autophagosomes. Consequently, the model in Figure 6I has to be corrected as not supported by the data and thus misleading. What is misleading is that the authors are throughout the manuscript talking about secretory autophagosomes, and in the model a "secretory amphisome" appears. This is a new concept for this reviewer, and the authors need to

explain it. How they distinguished in their experiment secretory autophagosomes from secretory amphisomes.

An important addition will be to show that non-lipidable LC3 is not recruited to the LC3-positive puncta, excluding a microtubule-related function of this protein in the secretion of the viral particles.

Figure 7e. The EM observation have to be quantified (number of autophagosomes per cell section, percentage of them containing viral particles....). This is particularly important because the authors infected cells at a MOI 10 for 24 h, and cells are very likely under stress. Stress induces autophagy and therefore the autophagosomes observed could be the result of this stress. As the cells are heavily infected, bulk autophagosomes will also enwrap some of the compartments containing the virions. This consideration is based on the morphology of the shown autophagosomes. Secretory autophagosomes carry a specific cargo in a selected manner. The autophagosomes presented in the figure of the authors mostly contain cytoplasm in their interior and therefore they are not selective. The way to accurately address this at the ultrastructural level (which this reviewer considers very important) is to infect cells with a lower MOI and examine cells by EM during the first replication cycle of Zika, when cells are not under a massive stress.

All the experiments carry out with viruses in Figure 6 and 7 are done with high MOI and/or for long times. Cells are very likely under stress and have been re-infected few times. The authors have to repeat some of their experiments by infecting cells with more modest MOI and look at their sample at the end of the first viral cycle of DENV and Zika (and not after 24 or 48 hours). If Lyn and secretory autophagosomes are a central factor for flavivirus viral cycle, findings will be recapitulated. This applies especially for figure 6h-6k.

What shown in figure 6l and 6m show that GRASP proteins are not involved in the release of VPLs and GRASP proteins are essential for secretory autophagy. How the authors interpret this?

If flaviviral particles are secreted via secretory autophagy, a population of this protein should be find associated with membranes containing the flaviviral particles.

Other comments

The authors mention in the introduction few papers about the possible involvement of autophagy in flavivirus infection, but there is also quite some substantial literature indicating that autophagy is not required for the viral life cycle of these viruses. This may also be indicated in the introduction, but certainly considered in the discussion while interpreting the data. How the authors explain those findings? Similarly, there are numerous ultrastructural studies on how flavivirus are secreted. How previous data fit with the findings of the authors (the authors just cite the two study that support their findings, but not other ones)?

Figure 3g. How do the authors know that the identity of the indicated band is indeed the one indicated?

Fig. 4k, 4l. The authors conclude that E protein are missorted to lysosomes in the absence of Lyn. Why those LAMP2-positive compartments are also LC3 positive?

Figure 5c. As the fractionation in this panel is compared with the one done with DENV-infected cells, the authors have to also assess the distribution of LC3.

Figure 6h. The authors state that the identified proteins in wild type and furin-sensitive VPL are known secretory autophagosome markers proteins, and for this they refer to one commentary and one

review, but not to an article with an effective biochemical characterization of those structures. This has to be corrected. Having said this, how the MS data were filtered/analyzed? Are the proteins presented in the panel the only ones detected? No viral particle component detected? Moreover, what is the secretory amphisome machinery?

Figure 7g. Please add the references demonstrating that the depleted proteins are marker or not of the secretory autophagosomes.

Response to reviewer 4

Reviewer: This is a much improved version of the manuscript previously submitted to Nat Microbiol. The data about the involvement of Lyn in the secretion of flavivirus are very solid, and experiments are very well performed and controlled.

The involvement of secretory autophagosomes, however, is more circumstantial and the authors do not prove this point directly. Therefore, the conclusions regarding how flavivirus are secreted have to be played down and be more balanced.

Response: In line with the reviewer's suggestion, we have changed the title and parts of the text in the manuscript (highlighted) to provide a more balanced conclusion and emphasised that secretory autophagy is one of the possibilities in virus secretion.

Major criticisms:

Figures 6i-6k. These experiments have also to be repeated with DENV and ZIKA.

Response: We have added data for Dengue and Zika infection (Figure S7a-d), as per the reviewer's suggestion, which are in agreement with our conclusions using VLPs.

Reviewer: Figures 6j/6k. At this point of the manuscript, the authors have no prove that the yellow puncta are autophagosomes (even later). LAPosomes (or any other cytoplasmic LC3-positive structure) are also yellow when labelled with RFP-GFP-LC3 and as a matter of fact, structures recruiting non-lipidated LC3 appear as yellow puncta as well. Overall, the authors cannot conclude that the yellow puncta that they are detecting at are secretory autophagosomes. Consequently, the model in Figure 6l has to be corrected as not supported by the data and thus misleading. What is misleading is that the authors are throughout the manuscript talking about secretory autophagosomes, and in the model a "secretory amphisome" appears. This is a new concept for this reviewer, and the authors need to explain it. How they distinguished in their experiment secretory autophagosomes from secretory amphisomes.

An important addition will be to show that non-lipidable LC3 is not recruited to the LC3-positive puncta, excluding a microtubule-related function of this protein in the secretion of the viral particles.

Response: We thank the reviewer for raising this point. LAPosomes are LC3+ phagosomal membranes most often observed during uptake of bacteria, fungi and parasites that rapidly fuse with lysosomes. Although they are LC3+, these compartments are acidic and therefore appear as "red" rather than "yellow" punctae, since GFP is sensitive to acidic pH (GFP pKa>6). Given that we see appearance of our LC3+ organelles with VLP-secretion alone (where there is no phagocytosis), combined with the genetic validation (depletion of genes of the secretory autophagosomal pathway), we believe secretory autophagosomes to be the most likely pathway. The model therefore is not based solely on the yellow punctae, but also on the mass spectrometry data, shRNA-based gene depletions of secretory autophagy and their effect on secretion of VLPs and infectious virus. To verify that the LC3 being recruited to these membranes are lipidated, we have treated cells with saponin to remove non-lipidated LC3 (Figure S7b). Our results confirm that the majority of LC3 in these non-degradative autophagosomes are lipidated.

Amphisomes are generated upon fusion of autophagosomes with endosomes, and are intermediate organelles often produced from these autophagosomes. Since we do see certain endosome markers in our virion enriched fractions (e.g. Rab11, Tfr), we cannot distinguish between autophagosomes and amphisomes. We have clarified this point in the revised version of the manuscript.

Reviewer: Figure 7e. The EM observation have to be quantified (number of autophagosomes per cell section, percentage of them containing viral particles....). This is particularly important

because the authors infected cells at a MOI 10 for 24 h, and cells are very likely under stress. Stress induces autophagy and therefore the autophagosomes observed could be the result of this stress. As the cells are heavily infected, bulk autophagosomes will also enwrap some of the compartments containing the virions. This consideration is based on the morphology of the shown autophagosomes. Secretory autophagosomes carry a specific cargo in a selected manner. The autophagosomes presented in the figure of the authors mostly contain cytoplasm in their interior and therefore they are not selective. The way to accurately address this at the ultrastructural level (which this reviewer considers very important) is to infect cells with a lower MOI and examine cells by EM during the first replication cycle of Zika, when cells are not under a massive stress.

Response: We thank the reviewer for raising this point. To clarify, in the infection conditions we had used, the cells were actually inoculated for 1.5 hours (not 24h) at MOI 10 and then washed and incubated for 24 h. These conditions that we previously selected for EM were based on conditions reported in the literature^{1,2}. We have however, re-performed these experiments at a lower MOI (2, 18h) which is the time-course of Zika life-cycle, to confirm the EM findings. We have added quantitation of total no of DMVs containing virus particles as suggested by the reviewer. With any shorter time interval or lower MOI it becomes technically difficult to locate progeny virions within cells.

Reviewer: All the experiments carry out with viruses in Figure 6 and 7 are done with high MOI and/or for long times. Cells are very likely under stress and have been re-infected few times. The authors have to repeat some of their experiments by infecting cells with more modest MOI and look at their sample at the end of the first viral cycle of DENV and Zika (and not after 24 or 48 hours). If Lyn and secretory autophagosomes are a central factor for flavivirus viral cycle, findings will be recapitulated. This applies especially for figure 6i-6k.

Response: As mentioned above, we have used conditions (MOI 2; 24 h infection) using conditions described in the literature², which results in ~70-80% of the population being infected. A single life cycle of Dengue/Zika takes ~18h; so data at 24h time point is essentially from a single round of infection. We have however added data for Dengue at lower MOI and shorter timepoints (MOI 1; 8, 16 and 24 hours post infection) to confirm the involvement of Lyn and the other genes of secretory autophagy in virus production (Fig S6a-e).

What shown in figure 6l and 6m show that GRASP proteins are not involved in the release of VLPs and GRASP proteins are essential for secretory autophagy. How the authors interpret this?

Response: We see an effect on secretion of infectious virus specifically with GRASP55 (~2 Log reduction) with the plaque assay but not with GRASP65. It is possible that the deficiency of GRASP65 results in compensatory effect from GRASP55. Dependence of unconventional secretion processes specifically on GRASP55, but not GRASP65 has also been observed with other cargo in primary mouse neurons and IL-1 β secretion from macrophages^{3,4}. Similarly, for the VLPs, we expect that compensatory effects in the individual depletions result in the lack of any significant defect, since a combined depletion of GRASP55 and 65 arrests secretion of the VLPs (data not shown). We have included text in the revised version of the manuscript to clarify this (highlighted).

Reviewer: If flaviviral particles are secreted via secretory autophagy, a population of this protein should be found associated with membranes containing the flaviviral particles.

Response: We have indeed detected markers of autophagosomes, in particular LC3 in the membrane bound fraction of flavivirus particles. We have added these data in the revised version of the manuscript (Fig S7e).

Other comments

Reviewer: The authors mention in the introduction few papers about the possible involvement of autophagy in flavivirus infection, but there is also quite some substantial literature indicating that autophagy is not required for the viral life cycle of these viruses. This may also be indicated in the introduction, but certainly considered in the discussion while interpreting the data. How the authors explain those findings? Similarly, there are numerous ultrastructural studies on how flavivirus are secreted. How previous data fit with the findings of the authors (the authors just cite the two study that support their findings, but not other ones)?

Response: The involvement of autophagy in the lifecycle of flaviviruses, in particular for their replication and assembly, has been reported extensively for multiple members of this family including Dengue, Zika, HCV, Japanese Encephalitis virus (JEV). An important exception is West Nile Virus infection where available data suggest that autophagy does not play a significant role⁵. The other contradictory data on a specific pro-viral effect of autophagy in Dengue and Zika infections is in mosquito cells where the data was inconclusive and suggested a limited role⁶. Since the primary focus of our study was on how viral progenies are *secreted* rather than *autophagy*, we had not included citations on autophagy. We have now included these references in the discussion section in the revised manuscript as per the reviewer's suggestion. Similarly for ultrastructural studies, available data have mostly focused on viral replication organelles rather than how they are secreted. We have included the relevant citations of these studies in the revised manuscript.

Reviewer: Figure 3g. How do the authors know that the identity of the indicated band is indeed the one indicated?

Response: The bands have been verified to be the ones indicated by immunoblotting and mass spectrometry in our previous studies⁷.

Reviewer: Fig. 4k, 4l. The authors conclude that E protein are missorted to lysosomes in the absence of Lyn. Why those LAMP2-positive compartments are also LC3 positive?

Response: In the microscopy images for both Dengue and Zika virus infected cells, only the Lyn KO cells have significant overlap of LAMP2 and LC3 markers. This is anticipated since a significant fraction of autolysosomes generated upon fusion of the virion-containing autophagosomes with lysosomes (in the absence of Lyn) are expected to be both LC3+ and LAMP2+ as also reported previously.

Reviewer: Figure 5c. As the fractionation in this panel is compared with the one done with DENV-infected cells, the authors have to also assess the distribution of LC3.

Response: We have included the distribution of LC3 in Fig 5c of the revised manuscript as suggested by the reviewer. The amount of detectable LC3 is very low in mock-infected cells due to basal autophagy levels, which is why we had excluded it from the previous version.

Reviewer: Figure 6h. The authors state that the identified proteins in wild type and furin-sensitive VPL are known secretory autophagosome markers proteins, and for this they refer to one commentary and one review, but not to an article with an effective biochemical characterization of those structures. This has to be corrected. Having said this, how the MS

data were filtered/analyzed? Are the proteins presented in the panel the only ones detected? No viral particle component detected? Moreover, what is the secretory amphisome machinery?

Response: We have included the available literature on characterization of this pathway as per the reviewer's suggestion. The mass spectrometry was performed on VLP-enriched fractions; so the viral prME protein served as the positive control and was indeed detected by MS. The criteria for inclusion in the dataset was identification of candidates in three independent runs. The criteria for selecting candidates from the mass spectrometry dataset was identification of at least 2 unique peptides. The proteins presented in the panel were those that were considered statistically significant after imposing a threshold of $\text{Log}_2[\text{fold enrichment}] > 4$ between control and sample sets. We have highlighted these details in the material/methods section to clarify this.

Amphisomes are intermediate organelles formed upon fusion of autophagosomes with that of early or late endosomes, which are frequent events as reported previously^{8,9}. Since our dataset includes candidates such as Rab11 and transferrin receptor, we cannot exclude amphisomes from autophagosomes. We have therefore referred to this pathway as autophagy-derived rather than secretory autophagy.

Reviewer: Figure 7g. Please add the references demonstrating that the depleted proteins are marker or not of the secretory autophagosomes.

Response: We have added the relevant literature in the revised manuscript.

References

1. Hamel, R. *et al.* Biology of Zika Virus Infection in Human Skin Cells. *J. Virol.* **89**, 8880–8896 (2015).
2. Welsch, S. *et al.* Composition and three-dimensional architecture of the dengue virus replication and assembly sites. *Cell Host Microbe* **5**, 365–375 (2009).
3. Chiritoiu, M., Brouwers, N., Turacchio, G., Pirozzi, M. & Malhotra, V. GRASP55 and UPR Control Interleukin-1 β Aggregation and Secretion. *Dev. Cell* **49**, 145-155.e4 (2019).
4. van Ziel, A. M., Largo-Barrientos, P., Wolzak, K., Verhage, M. & Scheper, W. Unconventional secretion factor GRASP55 is increased by pharmacological unfolded protein response inducers in neurons. *Sci Rep* **9**, 1567 (2019).
5. Beatman, E. *et al.* West Nile virus growth is independent of autophagy activation. *Virology* **433**, 262–272 (2012).
6. Brackney, D. E., Correa, M. A. & Cozens, D. W. The impact of autophagy on arbovirus infection of mosquito cells. *PLoS Negl Trop Dis* **14**, e0007754 (2020).
7. Zhang, J. *et al.* Flaviviruses Exploit the Lipid Droplet Protein AUP1 to Trigger Lipophagy and Drive Virus Production. *Cell Host Microbe* **23**, 819-831.e5 (2018).
8. Bader, C. A., Shandala, T., Ng, Y. S., Johnson, I. R. D. & Brooks, D. A. Atg9 is required for intraluminal vesicles in amphisomes and autolysosomes. *Biol Open* **4**, 1345–1355 (2015).
9. Berg, T. O., Fengsrud, M., Strømhaug, P. E., Berg, T. & Seglen, P. O. Isolation and characterization of rat liver amphisomes. Evidence for fusion of autophagosomes with both early and late endosomes. *J. Biol. Chem.* **273**, 21883–21892 (1998).

Reviewers' Comments:

Reviewer #4:

Remarks to the Author:

The authors have addressed sufficiently all the major criticisms. In their rebuttal letter, however, there are a couple of incorrect assertions. I point them out because grabbing the right concepts could be important for the interpretation in future studies.

Response: We thank the reviewer for raising this point. LAPosomes are LC3+ phagosomal membranes most often observed during uptake of bacteria, fungi and parasites that rapidly fuse with lysosomes. Although they are LC3+, these compartments are acidic and therefore appear as "red" rather than "yellow" punctae, since GFP is sensitive to acidic pH (GFP pKa>6).

>>> In LAPosomes but also in other situations in which LC3 is conjugated to the membrane of endosomes, LC3 is oriented cytoplasmically and not lumenally. As a result, LC3 cannot be "sensitive" to the endosomal pH and consequently, when the GFP-mRFP-LC3 probe is conjugated to endosomal membrane, the fluorescent signal is yellow AND NOT red.

Reviewer: Fig. 4k, 4I. The authors conclude that E protein are missorted to lysosomes in the absence of Lyn. Why those LAMP2-positive compartments are also LC3 positive?

Response: In the microscopy images for both Dengue and Zika virus infected cells, only the Lyn KO cells have significant overlap of LAMP2 and LC3 markers. This is anticipated since a significant fraction of autolysosomes generated upon fusion of the virion-containing autophagosomes with lysosomes (in the absence of Lyn) are expected to be both LC3+ and LAMP2+ as also reported previously.

>>> LC3 is rapidly degraded in lysosomes/autolysosomes, and therefore those are LAMP2+ and LC3-. The only way to have lysosomes/autolysosomes that are both LAMP2+ and LC3+ is when their degradative activity is inhibited!

Response to reviewer comments

Reviewer #4 (Remarks to the Author):

Reviewer: The authors have addressed sufficiently all the major criticisms. In their rebuttal letter, however, there are a couple of incorrect assertions. I point them out because grabbing the right concepts could be important for the interpretation in future studies.

In LAPosomes but also in other situations in which LC3 is conjugated to the membrane of endosomes, LC3 is oriented cytoplasmically and not lumenally. As a result, LC3 cannot be “sensitive” to the endosomal pH and consequently, when the GFP-mRFP-LC3 probe is conjugated to endosomal membrane, the fluorescent signal is yellow AND NOT red.

Response: We agree with the reviewer. However, there is no phagocytosis in the VLP-secreting cells and therefore these cells are very unlikely to produce LAPosomes.

Reviewer: Fig. 4k, 4l. The authors conclude that E protein are missorted to lysosomes in the absence of Lyn. Why those LAMP2-positive compartments are also LC3 positive? LC3 is rapidly degraded in lysosomes/autolysosomes, and therefore those are LAMP2+ and LC3-. The only way to have lysosomes/autolysosomes that are both LAMP2+ and LC3+ is when their degradative activity is inhibited.

Response: We agree with the reviewer. We can visualise LC3+ signal in the LAMP2+ vesicle at the 24h time point post infection, but much less LC3+ at a later time point in infection, which we presume is because of degradation as would be expected.